# The Eyes Absent family members EYA4 and EYA1 promote PLK1 activation and successful mitosis through tyrosine dephosphorylation

Christopher B. Nelson[1], Samuel Rogers[1], Kaushik Roychoudhury[2], Yaw Sing Tan[3], Caroline J. Atkinson[4], Alexander P. Sobinoff [1], Christopher G. Tomlinson[1], Anton Hsu[1], Robert Lu[1], Eloise Dray [5], Michelle Haber [4], Jamie I. Fletcher [4], Anthony J. Cesare[1], Rashmi S. Hegde [2] & Hilda A. Pickett [1] ✉

The Eyes Absent proteins (EYA1-4) are a biochemically unique group of tyrosine phosphatases known to be tumour-promoting across a range of cancer types. To date, the targets of EYA phosphatase activity remain largely uncharacterised. Here, we identify Polo-like kinase 1 (PLK1) as an interactor and phosphatase substrate of EYA4 and EYA1, with pY445 on PLK1 being the primary target site. Dephosphorylation of pY445 in the G2 phase of the cell cycle is required for centrosome maturation, PLK1 localization to centrosomes, and polo-box domain (PBD) dependent interactions between PLK1 and PLK1-activation complexes. Molecular dynamics simulations support the rationale that pY445 confers a structural impairment to PBD-substrate interactions that is relieved by EYA-mediated dephosphorylation. Depletion of EYA4 or EYA1, or chemical inhibition of EYA phosphatase activity, dramatically reduces PLK1 activation, causing mitotic defects and cell death. Overall, we have characterized a phosphotyrosine signalling network governing PLK1 and mitosis.

The Eyes Absent Family (EYA1-4) is a unique group of dual-function proteins with oncogenic roles in a variety of tumour types, where they promote cancer cell phenotypes such as proliferation, survival, and migration[1–9]. EYA proteins possess N-terminal transcriptional coactivation activity and C-terminal Haloacid Dehydrogenase (HAD) protein tyrosine phosphatase activity[5,10–19]. EYAs are the only known HAD-family tyrosine phosphatases and have unique active site chemistry making them well suited to specific inhibition with small molecules[20–23]. This has led to the identification of EYA phosphatase inhibitors, and there is growing interest in their chemotherapeutic potential[20,24–27]. However, the molecular functions and specific targets of EYA phosphatase activity are largely unknown. This is particularly the case for EYA4, which currently has no known phosphatase substrates.

Here, we employ an unbiased proteomics approach to identify EYA4 substrates that both interact with EYA4 and have tyrosine phosphorylation levels that respond to genetic or pharmacological perturbation of EYA4. We identify the essential mitotic regulator PLK1, and specifically Y445, a residue within polo-box 1 of PLK1, as a bona-fide substrate of both EYA4 and EYA1. EYA4 and PLK1 interact and colocalize specifically at centrosomes in G2 cells, and this interaction is dependent on a conserved putative polo-docking site (PDS) present on

---

[1]Children's Medical Research Institute, Faculty of Medicine and Health, University of Sydney, Westmead, NSW, Australia. [2]Division of Developmental Biology, Cincinnati Children's Hospital Medical Center, Cincinnati, OH, USA. [3]Bioinformatics Institute, Agency for Science, Technology and Research (A*STAR), Singapore, Singapore. [4]Children's Cancer Institute, Lowy Cancer Research Centre, School of Clinical Medicine, UNSW Medicine & Health, UNSW Sydney, Kensington, NSW, Australia. [5]University of Texas Health Science Center at San Antonio, San Antonio, TX, USA. ✉e-mail: hpickett@cmri.org.au

EYA4 and EYA1. We demonstrate that dephosphorylation of pY445 by EYA4 and/or EYA1 promotes PLK1 localization to centrosomes and centrosome maturation, normal spindle morphology, PLK1 PBD-dependent interactions, and PLK1 kinase activation. This mechanism is supported by structural simulations, which predict that Y445 phosphorylation reduces substrate recognition and alters polo-box function through flexibility changes within the connecting loop. Finally, we show that EYA-mediated dephosphorylation of PLK1 supports PLK1 function during mitotic progression, while treatment with an EYA phosphatase inhibitor potently elicits cell death in tumour cells expressing EYA4 and/or EYA1.

## Results

### EYA4 interacts with and dephosphorylates PLK1

To identify EYA4 substrates, we fused full-length EYA4 to a Myc-tagged BioID2 biotin ligase and identified EYA4 interacting proteins using a BioID2 proximity proteomics strategy[28,29] (Fig. 1A–B, Supplementary Fig. 1A–C). Biotinylation of proteins after expression of Myc-BioID2-EYA4 was compared to a BioID2-Myc control using mass spectrometry with label-free quantification (LFQ). This yielded 156 high-confidence EYA4 interactors (> 4-fold enrichment, $p$-value $\leq 0.05$) (Fig. 1A, Supplementary Data 1). Polo-like kinase 1 (PLK1) and Aurora kinase A (AURKA), two of the master regulatory kinases that control mitosis, were among the most highly significant EYA4 interacting proteins (Fig. 1A, Supplementary Data 1). PLK1 and AURKA localize to mitotic structures including the centrosome and spindle midbody, with PLK1 also localizing to the spindle and kinetochore[30]. Gene ontology analysis of our 156 high-confidence EYA4 interactors using STRING further supported a centrosomal role for EYA4, with 4 out of 28 significantly associated cellular component terms relating to the centrosome (spindle pole, centriole, centrosome, microtubule organizing center) (Supplementary Data 1). To further assess EYA4 interactors with relationships to mitosis and localization to mitotic structures we searched all EYA4 interactors with greater than 2-fold enrichment on pubmed and uniprot. We identified 54 EYA4 interactors with functional relationships to the centrosome, spindle, or mitosis. Additionally, 24 of these proteins have established localization to mitotic structures, including 17 with centrosomal localization (Fig. 1B, Supplementary Data 1).

To determine candidate EYA4 tyrosine phosphatase substrates, we depleted cells of EYA4 and performed immunoprecipitions using an anti-phosphotyrosine antibody followed by LC-MS/MS analysis. We identified 50 proteins with statistically significant increases in tyrosine phosphorylation, including PLK1 ($p \leq 0.05$, Supplementary Fig. 1D, E, Supplementary Data 1). Western blots of anti-phosphotyrosine immunoprecipitations confirmed an increase in PLK1 tyrosine phosphorylation in both 293 T and HeLa cells following EYA4 depletion (Fig. 1C). Using a reversed experimental design, we immunopurified endogenous PLK1 from cells arrested in G2 when PLK1 is concentrated at centrosomes, or following release into M-phase, when PLK1 becomes primarily localized to kinetochores, and blotted for tyrosine phosphorylation[31]. EYA4 depletion resulted in a striking increase in PLK1 tyrosine phosphorylation in G2, but this increase did not persist into mitosis (Fig. 1D). This suggests that EYA4 may regulate PLK1 function at G2 centrosomes through dephosphorylation.

### EYA4 and EYA1 contain a putative polo-docking site (PDS) that confers interaction with PLK1 in G2

Most PLK1 interactors possess a consensus PDS motif containing an internal phosphorylation site[32,33]. We identified a potential conserved PDS within the N-terminal transactivation domains of both EYA4 and EYA1, matching 6 of 7 residues of the consensus PDS sequence (AA 123-129, Fig. 1E, Supplementary Fig. 1F). Phosphorylation of pS128, which falls within the PDS of EYA4, has previously been observed in a high throughput dataset, suggesting that the PDS is functional (Fig. 1E)[34].

We used co-immunoprecipitation to determine whether the putative PDS on EYA4, and specifically phosphorylation of S128, mediated the interaction with PLK1. Myc-tagged WT EYA4 strongly interacted with PLK1 in G2 arrested cells. This interaction was abrogated for EYA4 containing a S128A non-phosphorylatable mutation and enhanced for EYA4 with a S128D phosphomimetic mutation (Fig. 1F). Further, while both the WT and the S128D interactions with PLK1 were reduced upon entry into mitosis, the S128D-PLK1 interaction persisted to some extent (Fig. 1F). These results suggest that phosphorylation of S128 in the putative EYA4 PDS regulates a G2-specific interaction between EYA4 and PLK1.

Consistent with an EYA4-PLK1 interaction in G2, we found that endogenous EYA4 colocalized with PLK1 at centrosomes in G2 (Fig. 1G). Colocalization between EYA4 and PLK1 was strongest in G2 cells with centrosomal PLK1 foci that were near each other, consistent with a less-mature pre-mitotic state (Supplementary Fig. 1G). This suggests that EYA-mediated dephosphorylation of PLK1 in G2 may regulate PLK1 functions in centrosome biology and early mitosis.

Myc-tagged EYA1 also interacts with PLK1 in G2 arrested cells, while EYA3, which lacks an obvious PDS, does not (Supplementary Fig. 1H). Further, depletion of EYA1, or both EYA4 and EYA1, also resulted in an increase in PLK1 tyrosine phosphorylation (Supplementary Fig. 1I), suggesting that both EYA4 and EYA1 can dephosphorylate tyrosine residues on PLK1 during G2.

### EYA4 and EYA1 promote PLK1 activation

PLK1 kinase activation begins at centrosomes and in the cytoplasm during G2 via AURKA-mediated phosphorylation of PLK1 at T210, reaching peak activation upon mitotic entry[30,35–38]. We determined whether EYA4 and EYA1 alter PLK1 activation by assessing pT210 phosphorylation in mitotic cells from asynchronous cultures using immunofluorescence with two different antibodies against pT210. In both HeLa and 293 T cells, co-depletion of EYA4 and EYA1 caused a highly significant decrease in pT210 staining using either antibody (Fig. 2A, B, Supplementary Fig. 2A–E). Individual knockdowns of EYA4 or EYA1 reduced PLK1 activation to differing magnitudes, with EYA4 depletion reaching statistical significance in HeLa cells using either antibody, and EYA1 depletion reaching statistical significance in 293 T cells (Fig. 2A, Supplementary Fig. 2A, knockdown confirmation: Supplementary Fig. 2B). These results could not be attributed to reduced PLK1 levels (Supplementary Fig. 2A).

Combination staining of pT210 and total PLK1 within the same experiment showed that there were significant reductions in the ratio of pT210/PLK1 in individual mitotic cells following depletion of EYA4, EYA1, as well as an additive decrease in the co-depleted condition, suggesting that EYAs support the mitotic accumulation of pT210 on a per-molecule basis (Fig. 2C).

To confirm that the reduction in pT210 was impacting PLK1 activity, we used immunofluorescence to evaluate the PLK1 kinase substrates pS46 TCTP, pS133 Cyclin B, and pS198 CDC25C[39–41] in mitotic cells following depletion of EYA4. Depletion of EYA4 significantly reduced staining of pS46 TCTP and pS198 CDC25C, and reductions in pS133 Cyclin B approached statistical significance (Fig. 2D–G). Depletion of EYA4 also significantly decreased pT210 levels relative to total PLK1 in western blots performed on lysates from nocodazole arrested mitotic cells (Fig. 2H).

In further support of EYA4 directly regulating PLK1 activity, overexpression of the EYA4 S128A mutant, or a D375N mutant (Ydef) previously shown to inactivate EYA4 phosphatase activity, resulted in lower levels of pT210/total PLK1 in nocodazole arrested cells relative to overexpression of a S128D construct (Fig. 2I)[24]. Lower levels of activated PLK1 were also observed by immunofluorescence following S128A mutant overexpression compared to WT EYA4 overexpression in unarrested mitotic cells (Supplementary Fig. 3A). These results suggest that EYA4 mediated PLK1 dephosphorylation is required to

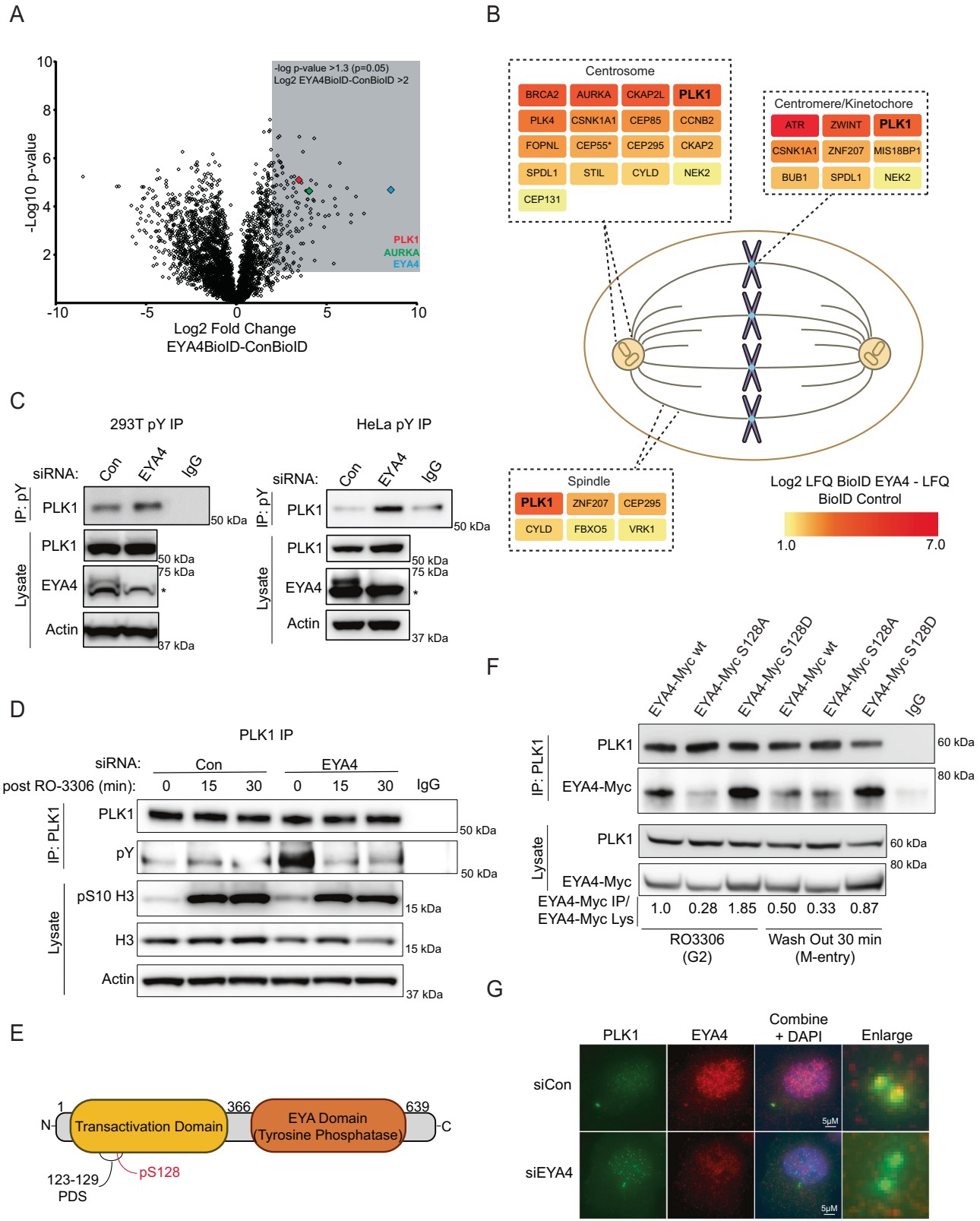

enhance PLK1 activation. We also observed a dose-dependent reduction of pT210 in mitotically arrested cells treated concurrently with the EYA phosphatase inhibitor benzarone and nocodazole for 12 h after thymidine release, reaching statistical significance in the 20 μM benzarone treatment (Fig. 2J). This coincided with a small reduction in PLK1 levels following benzarone treatment, such that the ratio of pT210 to total PLK1 was not significantly reduced (Fig. 2J).

## EYA4 and EYA1 support centrosome maturation, separation, and PLK1 localization to centrosomes

During G2 to M transition, PLK1 activation is essential for the accumulation of pericentriolar material at centrosomes (centrosome maturation), and for the separation of mature centrosomes[32,42–46]. To evaluate the effects of EYA4 and EYA1 depletion on centrosome maturation and separation, we measured both the number of

**Fig. 1 | EYA4 interacts with and dephosphorylates PLK1 in G2. A** Volcano plot of EYA4-Myc-BioID interactome. Grey coloured box represents 156 high confidence EYA4 interactors with -log p-value > 1.3 and Log2 fold difference > 2. EYA4, PLK1 and AURKA are indicated on the plot (n = 4, two-sided Student's *t*-test). **B** EYA4 interactors with known localization to mitotic structures including PLK1. Proteins have been colour coded by Log2 fold difference. **C** PLK1 tyrosine phosphorylation was assessed by western blots of eluates from immunoprecipitation (IP) of tyrosine phosphorylated proteins using an anti-phosphotyrosine antibody in 293 T or HeLa protein lysates following EYA4 depletion (n = 3 biological repeats). Asterisk represents non-specific band on the EYA4 blot. **D** Immunopurified endogenous PLK1 from control or EYA4 depleted 293 T cells that had been arrested in G2 with RO3306 or released into mitosis for 15 or 30 min were probed using an anti-phosphotyrosine antibody (pY, n = 1). **E** Schematic representation of EYA4 protein with N-terminal transactivation domain and C-terminal tyrosine phosphatase domain as well as putative PDS (AA 123- 129) and known phosphosite (pS128). **F** Co-immunopurification reactions between immunopurified endogenous PLK1 and an EYA4-Myc construct or EYA4-Myc mutants. Cells have been arrested in G2 with RO3306 or released into early M phase (n = 2 biological repeats). Densitometry of the relative EYA4-Myc or EYA4-Myc mutant in the IP eluate to that in the lysate is shown below the blot. **G** Immunoflourescence of endogenous PLK1 and EYA4 in G2 arrested HeLa cells treated with control siRNA or siRNA targeting EYA4. Enlarged images show closeups of PLK1 centrosomal foci demonstrating colocalization with EYA4 in the siCon treatment. Source data are provided as a Source data file.

---

centrosome foci and the level of pericentrin (PCNT) accumulated in centrosomal foci in prophase cells. Individual or combined depletion of EYA4 and EYA1 led to a significant increase in the proportion of prophase cells with only one centrosome focus (Fig. 3A, B). Prophase cells with only one centrosome focus also had lower centrosomal PCNT intensity following depletion of EYA4 or EYA1, with an additive effect being observed with combined depletion (Fig. 3C). These results support a role for EYA4 and EYA1 in centrosome maturation and separation.

PLK1 activation and centrosome maturation are intricately linked to PLK1 centrosomal localization in G2/M[30,36,38]. To evaluate the centrosomal localization of PLK1, asynchronous cells were pulse labelled with EdU, and G2/M cells were gated by their high DAPI intensity (equivalent to 4 N DNA content) and low EdU intensity. EYA4 depletion reduced the levels of PLK1 localization to centrosomes in G2/M cells without affecting the cytoplasmic/nuclear ratio of PLK1 (Fig. 3D–F, Supplementary Fig. 3B).

### The phosphatase activity of EYA4 and EYA1 prevents mitotic defects and mitotic cell death
We then examined HeLa cells for mitotic spindle defects following depletion of EYA4 and EYA1 or treatment with benzarone. Total spindle defects increased modestly following EYA4 or EYA1 depletion, and significantly following combined EYA4 and EYA1 depletion (Fig. 3G, H). Total spindle defects were highly significantly induced by treatment with benzarone (Fig. 3G, H). Monopolar spindles and spindles with misaligned chromosomes, two defects known to be caused by PLK1 deficiency, were frequently observed following combined depletion of EYA4 and EYA1, or benzarone treatment (Supplementary Fig. 3C)[47–50].

We next observed the effects of EYA4 and EYA phosphatase activity on mitotic duration and mitotic cell death. EYA4 depletion significantly prolonged mitosis and induced mitotic cell death (Fig. 3I, J). These defects were also observed following benzarone treatment (Fig. 3I, J). Further, benzarone caused a dose-dependent elevation in cleaved PARP, indicative of cell death, and phosphorylation of H3 on S10, indicative of mitotic arrest, in cancer cell lines with high expression levels of EYA4 (HeLa), EYA1 (SKNAS), or both (SKNFI) (Supplementary Fig. 3D, E).

A significant decrease in mitotic duration was also detected when we overexpressed the EYA4 S128D mutant that has enhanced interaction with PLK1 (Fig. 3K). Further, overexpression of the EYA4 Ydef mutant significantly increased mitotic death compared to the S128D mutant, supporting the rationale that overexpression of dysfunctional EYA4 can interfere with PLK1-mediated mitotic progression (Fig. 3L). These data demonstrate that EYA4 and EYA1 function to promote the accurate completion of mitosis.

### EYA4 targets pY445 on PLK1
To identify PLK1 tyrosine phosphosites that are targeted by EYA4, we first purified and trypsin-digested Myc-tagged PLK1 following knockdown of EYA4, treatment with benzarone, or the relevant control conditions (Fig. 4A). Next, we determined mass and charge data corresponding to potential phosphopeptides with untargeted mass spectrometry. Follow-up parallel reaction monitoring (PRM) mass spectrometry analysis confirmed the reliable detection of individual PLK1 phosphopeptides including pY445, pY425, pY421, and pY217 (Fig. 4B, Supplementary Fig. 4A–D).

Most of the PRM data could not be quantified in triplicate, due to one or more replicates having low dot product scores or an inconsistent number of quantifiable fragment ions. However, data corresponding to pY445 was of good quality across all replicates and was therefore quantified (Supplementary Fig. 4E, F). Although not reaching statistical significance, both knockdown of EYA4 and treatment with benzarone increased the pY445 peptide relative to the unmodified peptide, suggesting that pY445 may be a target of EYA4 (Supplementary Fig. 4E, F).

To determine if EYA4 has intrinsic biochemical specificity for pY445 or other PLK1 phosphosites, we performed in-vitro phosphatase assays using the purified phosphatase domain of EYA4 (residues 367-639) and synthetic phosphopeptides containing the four PLK1 tyrosine phosphosites (Fig. 4C). A scrambled version of pY445 was used as a negative control, and a peptide previously shown to be dephosphorylated by the phosphatase domains of all four EYA proteins was used as a positive control (H2AX pY142)[25,51]. Interestingly, all peptides were able to be dephosphorylated; however, the kinetics of dephosphorylation were fastest, and similar to the positive control peptide, for pY445 and pY425. pY445, pY425, and the pY142 H2AX positive control peptide were fit by Michaelis-Menten kinetics and had Km values of 0.46 mM, 0.34 mM, and 0.51 mM, respectively (Fig. 4C, Supplementary Fig. 4G). In contrast, pY217, pY421, and pY445-scrambled were dephosphorylated with slower kinetics and could not be fit with Michaelis-Menten kinetics at the concentrations tested (Fig. 4C, Supplementary Fig. 4G). These results suggest that the phosphatase domain of EYA4 has greater intrinsic specificity for pY425 and pY445.

To determine the contribution of pY445 dephosphorylation to the increased tyrosine phosphorylation of PLK1 observed upon EYA4 depletion, we compared tyrosine phosphorylation of an unphosphorylatable Y445F mutant with WT PLK1 following EYA4 depletion in G2 arrested cells. EYA4 depletion increased overall PLK1 tyrosine phosphorylation, however, no induction was observed for Y445F (Supplementary Fig. 5A). This implicates pY445 as the predominant PLK1 phosphosite targeted by EYA4.

The phosphatase domains of EYA1 and EYA3 were also tested for their ability to dephosphorylate the pY445 peptide. Both EYA1 and EYA3 were able to dephosphorylate pY445 with fast Michaelis-Menten kinetics (Supplementary Fig. 5B). These results suggest that dephosphorylation of PLK1 pY445 is regulated both by intrinsic specificity within the highly homologous EYA phosphatase domain (~70–90% homology between EYA members), as well as by the ability of EYA4 and EYA1 to interact with PLK1 in-vivo through a PDS.

### A non-phosphorylatable Y445F PLK1 mutant is hyperactive
We hypothesised that elevated phosphorylation of Y445 was responsible for the reduction of PLK1 activity observed following

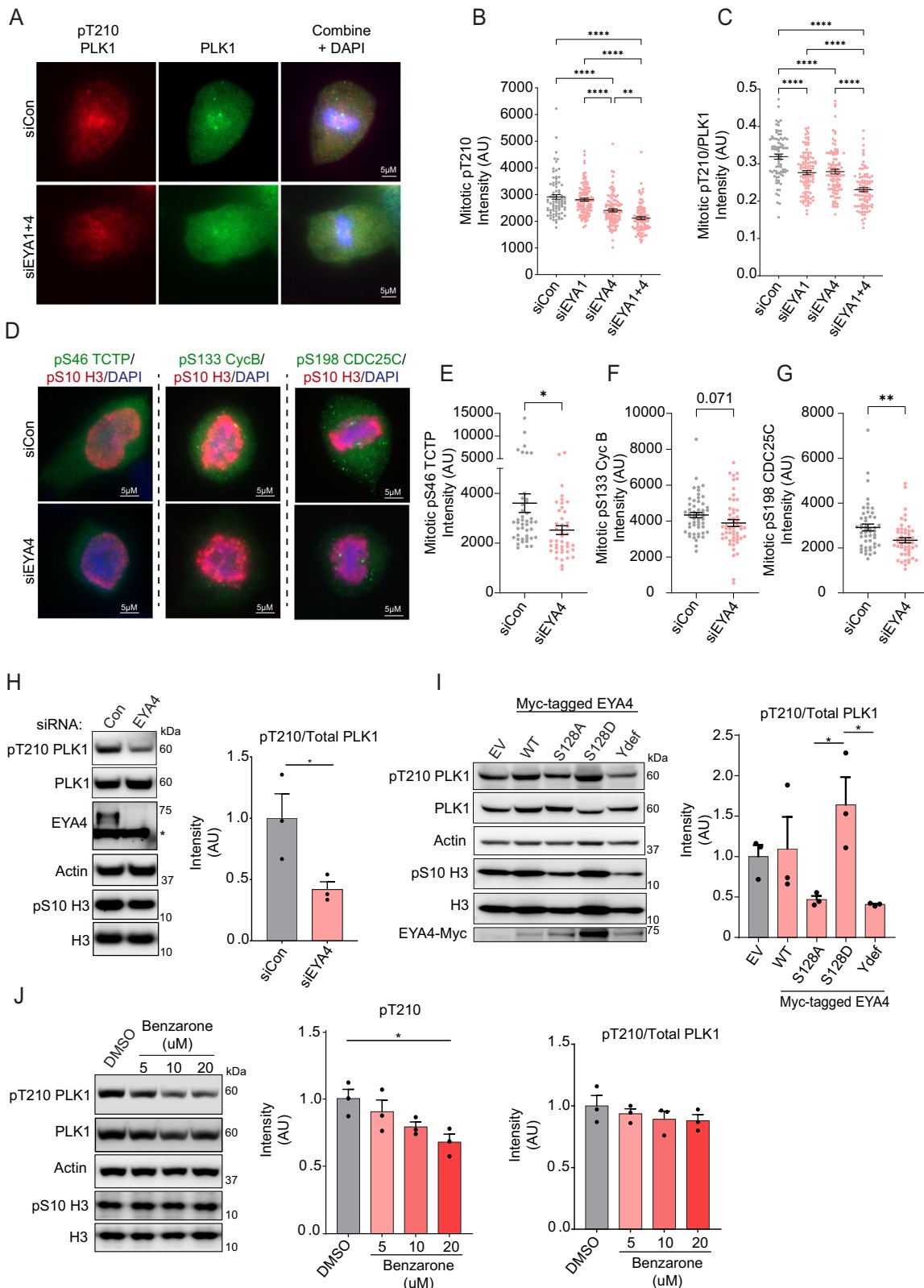

depletion of the EYAs. In support of this hypothesis, overexpressed PLK1-Y445F had elevated pT210 phosphorylation levels relative to WT PLK1 in mitotically arrested HeLa and 293 T cells (Fig. 5A, B, Supplementary Fig. 5C).

Our in-vitro phosphatase data suggested that EYA4 could potentially dephosphorylate pY425 as well as pY445. We therefore tested the effect of Y425F and a Y425F/Y445F double mutant on PLK1 activation.

Interestingly, neither Y425F nor the Y425F/Y445F double mutant significantly altered T210 phosphorylation, suggesting that pY445 dephosphorylation is responsible for EYA-mediated changes in PLK1 activity (Supplementary Fig. 5C). Additionally, these results suggest that the phosphorylation state of pY425 may oppose the inhibitory effects of pY445 on PLK1 activation, consistent with data showing that pY425 can increase PLK1 activity[52].

**Fig. 2 | EYA4 and EYA1 support PLK1 activation. A** Representative images of mitotic HeLa cells stained for total PLK1 and pT210 PLK1 following knockdowns. **B** Quantitation of cellular fluorescence intensity of pT210 PLK1 in mitotic cells following knockdowns (independent cells measured from left to right: 80, 102, 96, 89, **p = 0.0073, ****p ≤ 0.0001, ANOVA with Tukey correction). **C** Quantitation of cellular fluorescence intensity of pT210/total PLK1 mitotic cells following knockdowns (independent cells measured from left to right: 80, 102, 96, 89, ****p ≤ 0.0001, ANOVA with Tukey correction). **D** Representative images of individual mitotic HeLa cells stained for PLK1 substrates (green), pS10 H3 (red), and DAPI, with and without depletion of EYA4. **E** Quantitation of cellular fluorescence intensity of pS46 TCTP in mitotic cells (n = 43 cells per condition, *p = 0.0105. two-sided Student's t-test). **F** Quantitation of cellular fluorescence intensity of pS133 Cyclin B in mitotic cells (n = 53 cells per condition, p = 0.071, two-sided Student's t-test).

**G** Quantitation of cellular fluorescence intensity of pS198 CDC25C in mitotic cells (n = 49 cells per condition, **p = 0.0031, two-sided Student's t-test). **H** Western blots from nocodazole arrested HeLa cells and densitometry of pT210 PLK1/total PLK1 (n = 3 biological replicates, *p = 0.0495, two-sided Student's t-test, asterisk indicates a non-specific band). **I** Representative western blots and pT210/total PLK1 densitometry in nocodazole arrested HeLa cells overexpressing EYA4 or EYA4 mutants. (n = 3 biological replicates, *p = 0.0417 for comparison between S128A and S128D and 0.0318 for comparison between S128D and Ydef, ANOVA with Tukey correction). **J** Western blots following treatment with a pan-EYA phosphatase inhibitor in nocodazole arrested HeLa cells and densitometry of pT210 and pT210/total PLK1 (n = 3 biological replicates, *p = 0.0248, ANOVA with Dunnet correction). "AU" stands for arbitrary units. All quantitative data in this figure are presented as mean values +/- SEM. Source data are provided as a Source data file.

To explore the functional consequences of Y445 phosphorylation, we performed rescue experiments following depletion of endogenous PLK1 using a 3'UTR directed siRNA (Fig. 5C–F). Depletion of endogenous PLK1 caused a dramatic induction of cleaved PARP, indicative of cell death in cells expressing an empty vector (EV) construct (Fig. 5C, D). Cleaved PARP levels were significantly reduced by overexpression of WT PLK1 or Y445F PLK1, with the reduction being of greater magnitude for the latter (Fig. 5C, D). Expression of WT PLK1 and Y445F PLK1 increased following depletion of endogenous PLK1, suggesting that the surviving cells were those with exogenous expression (Fig. 5C).

Live cell imaging rescue experiments identified that WT PLK1 only slightly rescued the fraction of mitotic death events caused by endogenous PLK1 depletion (p = 0.06), while the surviving cells did have a significantly reduced mitotic duration (Fig. 5E, F). In contrast, Y445F PLK1 dramatically reduced the fraction of mitotic cell death events but did not significantly reduce mitotic duration in the surviving cells (Fig. 5E, F). These results suggest that exogenous myc-tagged PLK1 is somewhat less potent than endogenous PLK1, and that the reduced cleaved-PARP following rescue with WT PLK1 may be partially indicative of faster repopulation by surviving cells. Overall, Y445F more robustly prevents cell death, consistent with hyperactivation of Y445F.

As pY445 is a substrate of EYA4 we hypothesized that Y445F PLK1 overexpressing cells might be less sensitive to EYA4 depletion. Interestingly, overexpression of WT PLK1 itself reduced the effects of EYA4 depletion on mitotic cell death compared to previous experiments (Fig. 5G, Fig. 3J). However, even in conditions of WT PLK1 overexpression, EYA4 depletion still caused a significant increase in both mitotic cell death and mitotic duration (Fig. 5G, H). In comparison, there were no significant increases in mitotic death or duration following EYA4 depletion in cells overexpressing Y445F PLK1 (Fig. 5I, J). In addition to these relative effects, we also compared the absolute mitotic duration and fraction of mitotic cell death between WT PLK1 and Y445F PLK1 expressing cells following depletion of EYA4. While we did not observe a difference in the absolute level of mitotic cell death, we did find that EYA4 depletion resulted in a significantly longer mitotic duration in WT PLK1 overexpressing cells compared to Y445F PLK1 overexpressing cells.

To determine whether Y445 phosphorylation status impacts the cellular outcomes of EYA inhibition, we performed Alamar blue viability dose-response curves with benzarone in cells stably overexpressing an EV construct, WT PLK1, Y425F PLK1, or Y445F PLK1. Cell sensitivity was similar in EV and WT PLK1 overexpressing cells, but Y425F PLK1 expressing cells were slightly more sensitive to benzarone, and Y445F PLK1 expressing cells were significantly more resistant (Fig. 5K). These results are consistent with the conclusion that cell death in response to EYA inhibition is caused in part by the prevention of pY445 PLK1 dephosphorylation.

### Y445 phosphorylation inhibits the interaction with PLK1 activation complexes

Phosphorylation of PLK1 at pT210 by AURKA requires direct interaction between the PBD of PLK1 and a PDS on one of two AURKA cofactors:

BORA in the cytoplasm, or CEP192 at centrosomes[30]. As Y445 falls within the PBD of PLK1, and Y445F is hyperactive, we explored whether Y445 phosphorylation impacts the ability of PLK1 to interact with AURKA cofactors. We found that the Y445F mutant yielded a much stronger interaction with BORA compared to WT PLK1 in G2 and mitotically arrested cells (Fig. 6A). The interaction between Y445F and CEP192 was also increased relative to WT PLK1 specifically in mitotic cells (Fig. 6A). We observed a modest increase in the interaction between Y445F and AURKA itself in mitotic cells despite the transient nature of the PLK1-AURKA interaction (Fig. 6A). Interestingly, in mitotic cells, Y445F also interacted more strongly than WT PLK1 with BUB1, a major kinetochore receptor for PLK1 in mitosis that interacts with the PBD of PLK1 (Fig. 6A)[53,54]. These data suggest that Y445 phosphorylation reduces the affinity of the PLK1 PBD for PLK1 interaction partners, which underpins a reduction in PLK1 activity and functionality.

### Phosphorylation of Y445 is predicted to reduce flexibility within the PBD connecting loop and substrate binding

We next investigated whether Y445 phosphorylation disrupts the structure and function of the PLK1 PBD. We performed molecular dynamics (MD) simulations using a published structure of the PBD of PLK1 bound to a model phosphopeptide (PDB: 1UMW)[32]. The molecular mechanics/generalized Born surface area (MM/GBSA) method was used to compute the binding free energy of the protein-peptide interaction for the WT PLK1 PBD and PBD structures in which we simulated the phosphorylation of either Y445 or Y425[55]. While phosphorylation of Y425 did not alter the computed binding free energy, phosphorylation of Y445 resulted in a significantly higher binding free energy (Supplementary Data 2). When we compared our simulated PBD with phosphorylation of Y445 to a simulated mutant (Y445F), we again observed an increase in the binding free energy of the simulated phosphorylation. These results support our experimental data, in which the unphosphorylatable Y445F PLK1 mutant interacted more strongly with PLK1 activation complexes than WT PLK1 (Fig. 6A).

Additionally, pY445 is predicted to form intimate salt bridges with R507 (average occupancy of 96% and 88% in simulations of unbound pY445 PLK1 and bound pY445 PLK1, respectively) on the backside of the phosphopeptide binding pocket. This interaction forms part of a hydrogen bonding network that also includes D429 and N446 (Fig. 6B). Overall, the altered interactions caused by pY445 reduce the flexibility of the connecting loop between the two PBD domains, particularly at residues 502-506, which is a region normally associated with a high degree of disorder and is not resolved in several crystal structures (Fig. 6C, D, Supplementary Fig. 6A, B)[42,56–58]. Loss of flexibility in the pY445 simulation occurred irrespective of whether the PBD structure was bound to the model phosphopeptide (Fig. 6C, D, Supplementary Fig. 6A, B).

When PLK1 is in the nonphosphorylated state, R507 does not interact with Y445, but does interact intermittently with the nearby interdomain loop residues, E501, G502, D503, and E504. Most of these interactions are lost or significantly attenuated when Y445 is

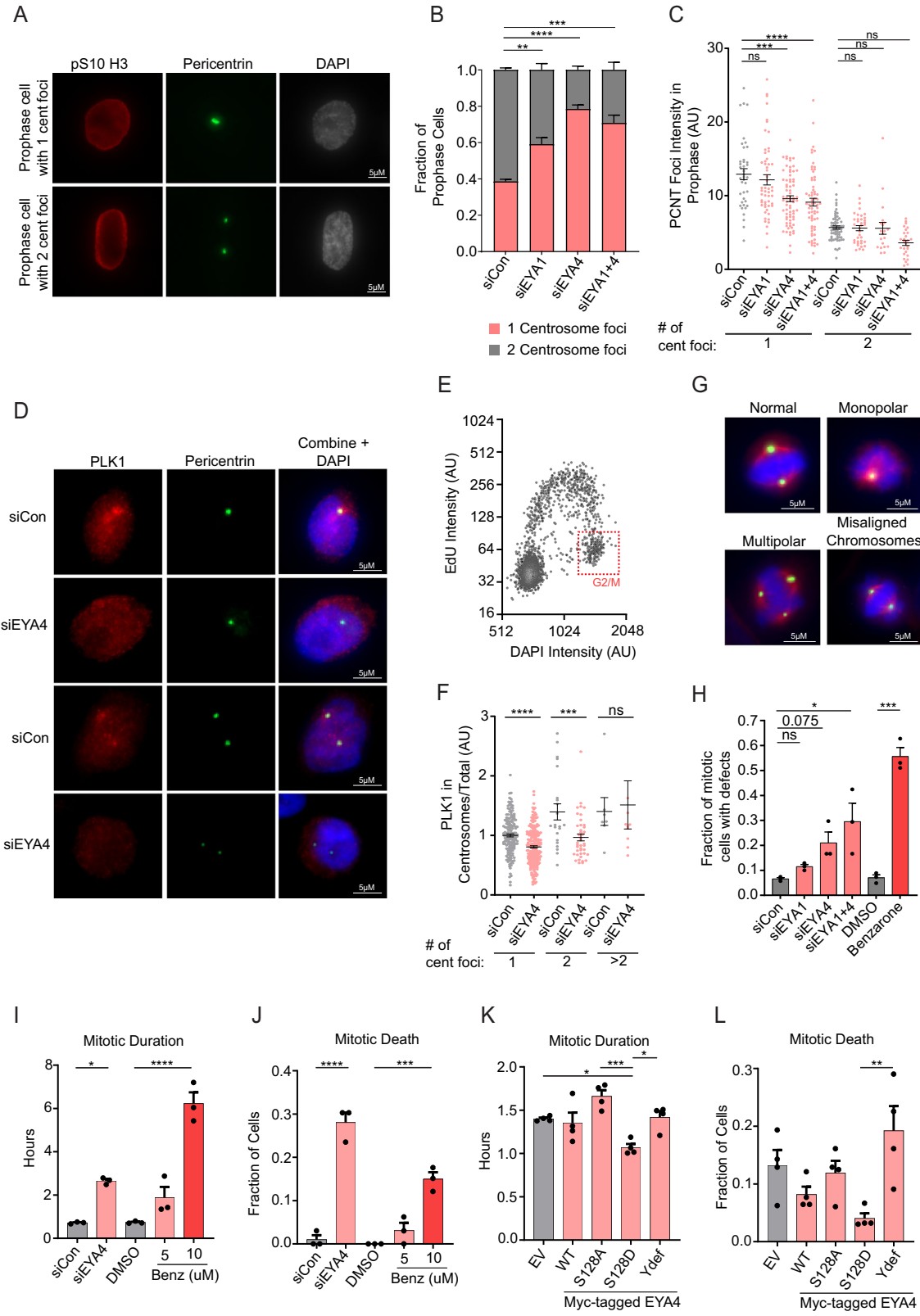

phosphorylated. These results suggest that phosphorylation of Y445 causes PBD dysfunction by perturbing both substrate binding and the overall structural dynamics of the PBD.

## Discussion
The EYA family is a biochemically unique group of protein tyrosine phosphatases; however, little is known about the substrates of the EYAs, or the downstream cellular processes impacted by EYA-mediated dephosphorylation. Using BioID proximity proteomics, we provide a comprehensive characterization of EYA4 binding partners and find that many EYA4 interactors are involved in the regulation of mitosis. We identify PLK1, an essential mitotic kinase, as an EYA4 phosphatase substrate, and describe a signalling pathway in which EYA4 and EYA1 regulate PLK1 through the dephosphorylation of pY445

**Fig. 3 | EYA4 and EYA1 support PLK1 functions in centrosome biology and mitosis. A** Prophase HeLa cells with 1 or 2 centrosome foci (pericentrin staining). **B** Quantitation of centrosome foci number following knockdowns ($n = 3$, **$p = 0.0061$, ***$p = 0.0003$, ****$p \leq 0.0001$, ANOVA with Tukey correction). **C** Integrated intensity of pericentrin foci following knockdowns (Independent cells from left to right: 103, 91, 93, 92, ***$p = 0.0001$, ****$p \leq 0.0001$, ANOVA with Tukey correction). **D** PLK1 colocalization with pericentrin in G2 cells. **E** G2 cells identified by gating of EdU and DAPI intensities. **F** Quantitation of PLK1 integrated intensity at centrosomes in G2 cells (Independent cells measured from left to right: 164, 232, 22, 41, 7, 10, ***$p = 0.0004$, ****$p \leq 0.0001$, ANOVA with Tukey correction). **G** Examples of spindle defects (Green: pericentrin, Red: alpha-tubulin). **H** Quantification of spindle defects ($n \geq 87$ mitotic cells across three experiments, Independent cells measured from left to right: 92, 87, 98, 103, 127, 147, *$p = 0.0238$, ***$p = 0.0002$,

siRNA experiment: ANOVA with Tukey correction, drug treatments: two-sided Student's *t*-test). **I** Mitotic duration following knockdowns or EYA inhibition ($n \geq 90$ across 3 experiments, Total number of independent cells measured from left to right: 99, 101, 100, 97, 100, *$p = 0.0122$, ****$p \leq 0.0001$, ANOVA with Tukey correction). **J** Mitotic cell death following knockdowns or EYA inhibition ($n \geq 90$ across 3 experiments, ***$p = 0.0003$, ****$p \leq 0.0001$, ANOVA with Tukey correction). **K** Mitotic duration following overexpression of EYA4 WT or mutants ($n \geq 100$ across 4 experiments, *$p = 0.0373$ between EV and S128D, $p = 0.0274$ between S128D and Ydef, ***$p = 0.0003$, ANOVA with Tukey correction). **L** Mitotic cell death following overexpression of EYA4 WT or mutants ($n \geq 100$ across 4 experiments, **$p = 0.0059$, ANOVA with Tukey correction). All quantitative data in this figure are presented as mean values +/− SEM. Source data are provided as a Source data file.

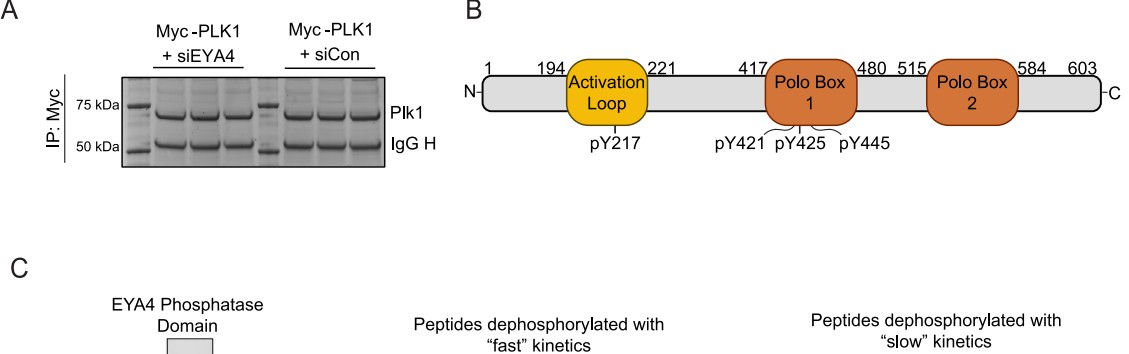

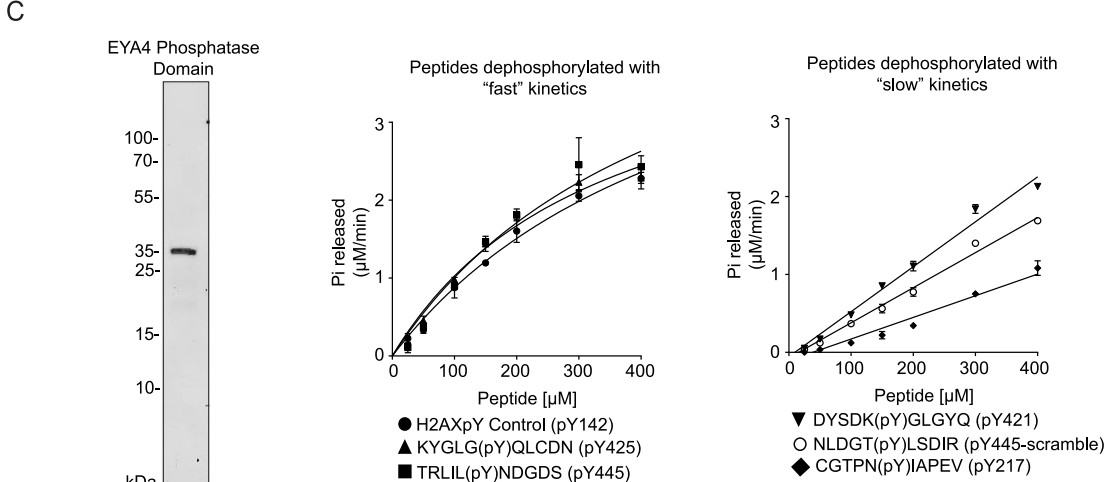

**Fig. 4 | PLK1 phosphotyrosine peptides and their in-vitro dephosphorylation by the EYA4 phosphatase domain. A** Schematic showing an example of immunopurified and Coomassie stained PLK1 which was in-gel digested and used in downstream pilot and PRM mass spec experiments. **B** PLK1 tyrosine phosphosites detected by mass spec with their positions designated on a PLK1 gene schematic. **C** The purified phosphatase domain of EYA4 (left, Coomassie-stained SDS-PAGE gel) was used in in-vitro phosphatase assays with synthetic phosphopeptides in

duplicate. Peptides dephosphorylated with relatively fast kinetics ($n = 2$, pY445, Km 0.46 mM; pY425, Km 0.34 mM; and a positive control peptide from H2AX pY142, 0.51 mM) were fit to Michaelis-Menten curves (middle panel). Peptides with slower kinetics, which could not be fit to a Michaelis-Menten curve at the substrate concentrations tested, were fit using linear regressions (right panel). Individual datapoints represent means. Error bars represent standard deviations. Source data are provided as a Source data file.

during G2 (Fig. 7). Dephosphorylation of PLK1 by the EYAs supports PLK1 activation through structural changes within the PBD, and impacts PLK1 regulated cellular processes such as centrosome maturation and separation, spindle formation, and mitotic progression. While it has previously been shown that Y445 can be phosphorylated by c-ABL, this is the first report of the impact of Y445 phosphorylation status on the structure and function of the PBD, as well as its role in regulating PLK1 activation[52].

While our overall conclusions are supported by the use of an EYA phosphatase inhibitor, there were some phenotypic differences observed when comparing the inhibitor to genetic depletion of the EYAs. In particular, treatment with benzarone reduced T210 phosphorylation as well as PLK1 levels in mitotic cells, compared to an exclusive reduction of T210 phosphorylation following knockdown.

Further, while both knockdowns and benzarone treatment caused an induction of mitotic defects, the magnitude of the induction was greater with benzarone treatment, and benzarone treatment tended to cause misaligned chromosomes to a greater proportion than other defect types. These differences may be attributed to potential off target effects of benzarone, or the effects of benzarone on EYA2 and EYA3. A recent paper reported that EYA2 can localize to centrosomes and support mitosis, even though it does not have an obvious PDS sequence[17]. Therefore, EYA2 may play a complementary role to EYA1 and EYA4, potentially having other substrates that impact upon mitotic signalling and PLK1.

We identified an unphosphorylatable Y445F mutant as hyperactive and more potent in supporting mitosis compared to WT PLK. Therefore, pY445 dephosphorylation provides a mechanistic

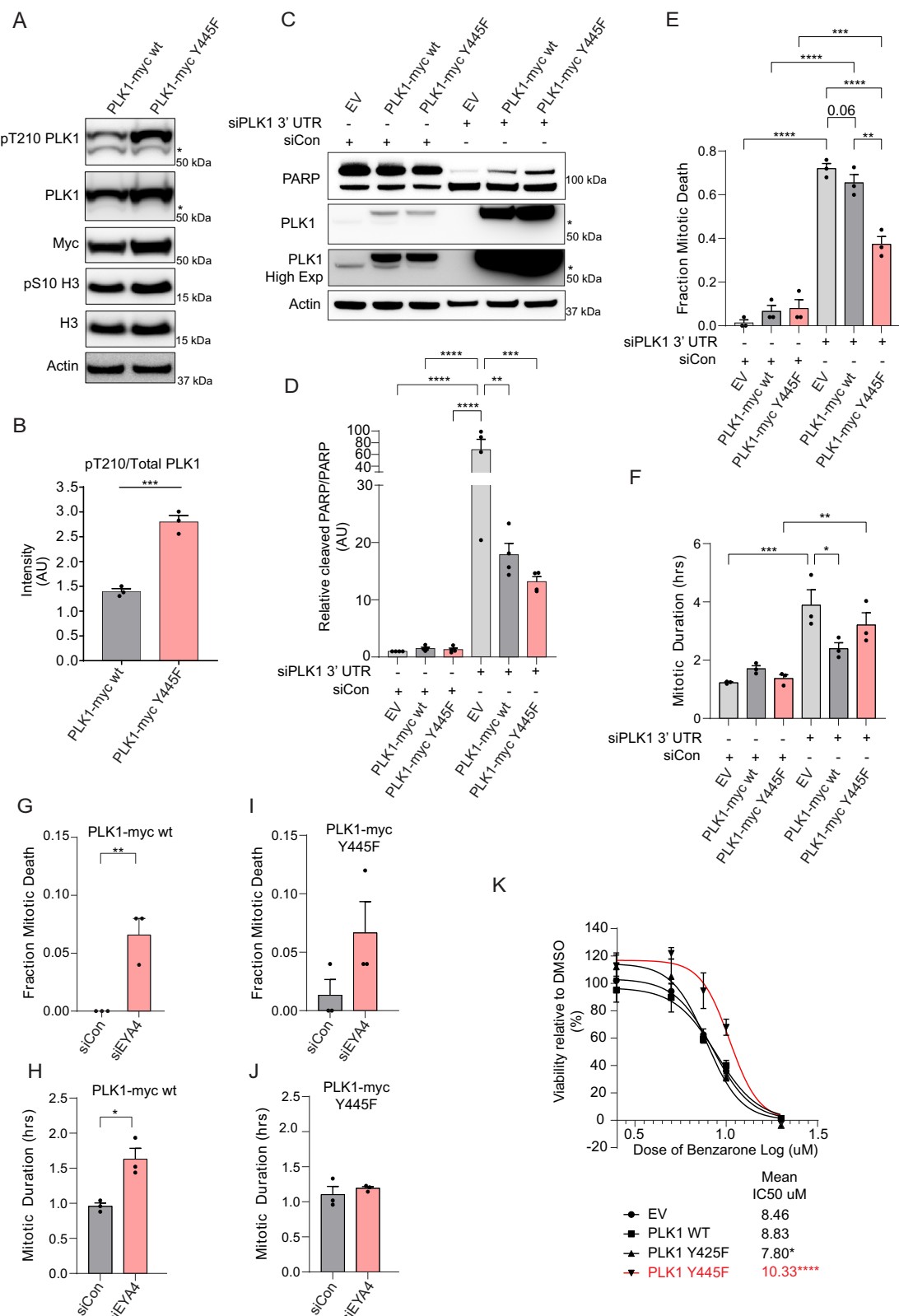

explanation for how EYAs support PLK1 activation and mitosis, although the involvement of additional PLK1 phosphosites cannot be entirely ruled out. A pY425 mutant did not affect PLK1 activity in our hands, suggesting that pY425 dephosphorylation does not contribute to the promotion of PLK1 activation by the EYAs. However, mutants of pY217 and pY421 were not tested. Exact Km values for pY217 and pY421 peptides could not be accurately determined by in-vitro phosphatase

assays due to substrate solubility issues at higher concentrations, but it is clear that these peptides have slower dephosphorylation kinetics than pY445, making them less likely to be EYA4 substrates. Additionally, the Y445F mutant did not exhibit increased tyrosine phosphorylation upon EYA4 depletion, suggesting that pY445 is primarily responsible for the induction of tyrosine phosphorylation observed when EYA4 is depleted in endogenous conditions or when

**Fig. 5 | PLK1 Y445F is hyperactive and provides greater resistance to effects on survival and mitosis caused by depletion of endogenous PLK1, EYA4, or EYA inhibition. A** Representative western blots of pT210 and total PLK1 in PLK1-myc WT and PLK1-myc Y445F expressing HeLa cells that have been mitotically synchronized with nocodazole (* represents endogenous pT210 or total PLK1). **B** Densitometry pertaining to (**A**) ($n = 3$ biological replicates, ***$p = 0.0006$, two-sided Student's $t$-test). **C** Representative western blots of endogenous PLK1 depletion rescue by PLK1-myc WT and PLK1-myc Y445F (* represents endogenous PLK1).
**D** Densitometry of cleaved PARP/PARP pertaining to (**A**) ($n = 4$, **$p = 0.0013$, ***$p = 0.0005$, ****$p \leq 0.0001$, ANOVA with Tukey correction for multiple comparisons). **E, F** Quantification of mitotic cell death (**E**) or mitotic duration (**F**) from live cell microscopy experiments involving endogenous PLK1 depletion and rescue with PLK1-myc WT or PLK1-myc Y445F overexpression ($n = 75$ mitotic entry events

across 3 experiments, pertaining to (**E**): **$p = 0.0011$, ***$p = 0.0001$, ****$p \leq 0.0001$, pertaining to (**F**): *$p = 0.0326$, **$p = 0.0080$, ***$p = 0.0004$, ****$p \leq 0.0001$, both panels used ANOVA with Tukey correction for multiple comparisons).
**G–J** Quantification of mitotic cell death (**G, I**) or mitotic duration (**H, J**) from live cell microscopy experiments involving depletion of EYA4, and overexpression of PLK1-myc WT or PLK1-myc Y445F overexpression ($n = 75$ mitotic entry events across 3 experiments, *$p = 0.0139$, **$p = 0.0075$). **K** Alamar blue viability dose-response curves following 72 h treatment with benzarone across a range of concentrations in cells stably overexpressing an EV construct, PLK1-myc WT, or PLK1-myc mutants ($n = 4$ replicates, mean IC50 values presented, *$p \leq 0.05$, ****$p \leq 0.0001$, two-sided Student's $t$-tests). "AU" stands for arbitrary units. All quantitative data in this figure are presented as mean values +/− SEM. Source data are provided as a Source data file.

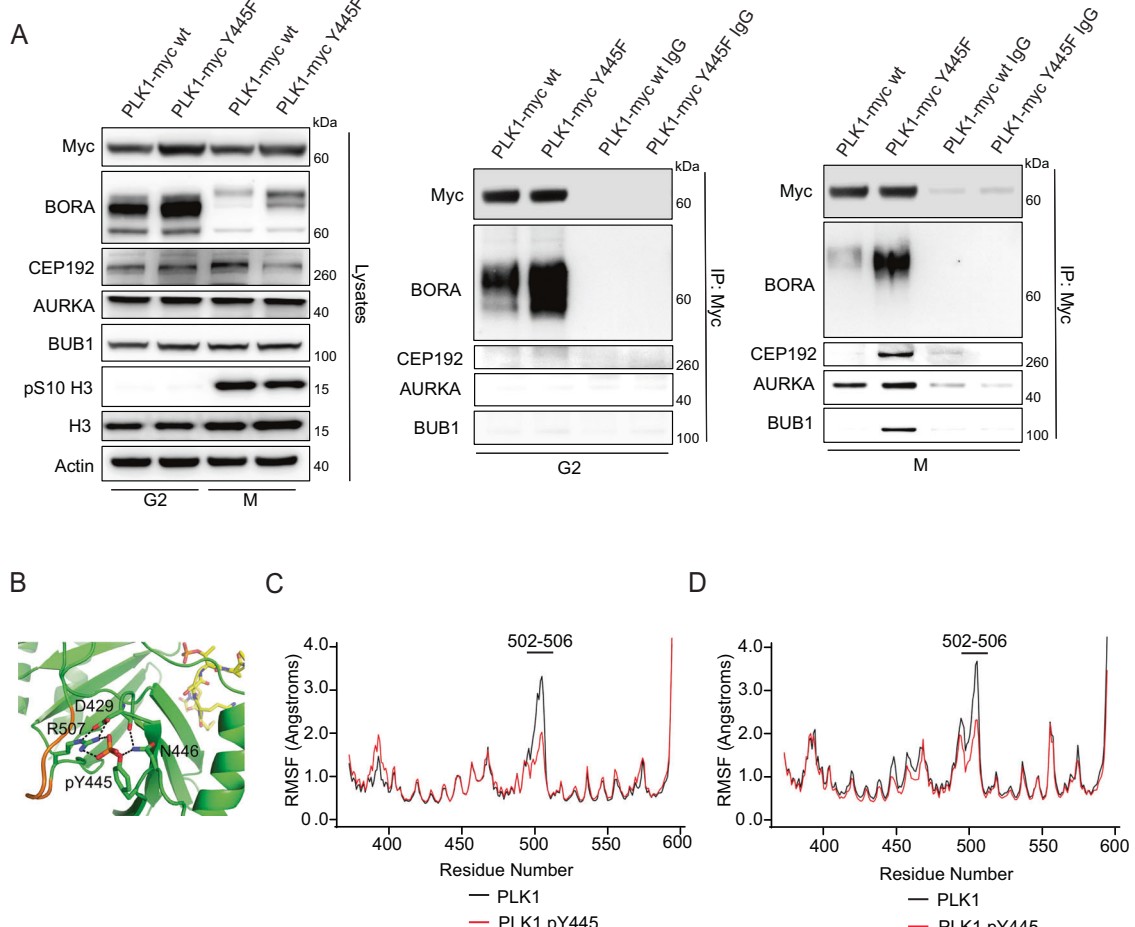

**Fig. 6 | pY445 alters the structure and function of the PBD. A** Co-immunoprecipitation reactions with overexpressed PLK1-myc WT or PLK1-myc Y445F in RO3306 arrested G2 cells and nocodazole arrested M-phase cells. Western blots from protein lysates (left panel), and western blots from G2 arrested eluates (middle panel) and M-phase arrested cells (right panel). **B** Representative simulation snapshot of the backside of the phosphopeptide binding pocket of PLK1s PBD (bound phosphopeptide in yellow). Y445 is indicated in a phosphorylated state (pY445). The image highlights the interaction of pY445 with R507 (green sticks), as

well as hydrogen bonds formed by D429 and N446, and increased rigidity within the connecting loop (AA 502-506) highlighted in orange. **C, D** Plots of the root mean square fluctuation (RMSF) of Cα atoms in the WT PLK1 PBD (black line) and Y445-phosphorylated PLK1 PBD (red line). The decreased RMSF within the connecting loop (AA 502-506) is indicative of decreased flexibility either when the PBD is bound to a model phosphopeptide (**C**), or in the unbound state (**D**) ($n = 5$ simulations). Source data are provided as a Source data file.

overexpressing WT PLK1. We conclude that exclusive pY445 dephosphorylation by the EYAs is the most parsimonious explanation for the observed effects on PLK1 and mitosis.

It has been shown that c-Abl can phosphorylate pY445, pY425 and pY217[52]. Given that two of these residues have now been shown to reduce PLK1 activity (pY217[59] and pY445), while one has been shown to promote PLK1 activity (pY425[52]), dynamic dephosphorylation is likely

to be crucial to the fine tuning of PLK1 activation. We observed that the activating effect of an unphosphorylatable Y445F mutation was reduced by a concomitant unphosphorylatable Y425F mutation, suggesting that individual phosphosites can also alter the functional consequences of each other. The interplay between different phosphosites, their impact on PLK1 activity, and which of these states are most biologically relevant will be an important area of future study.

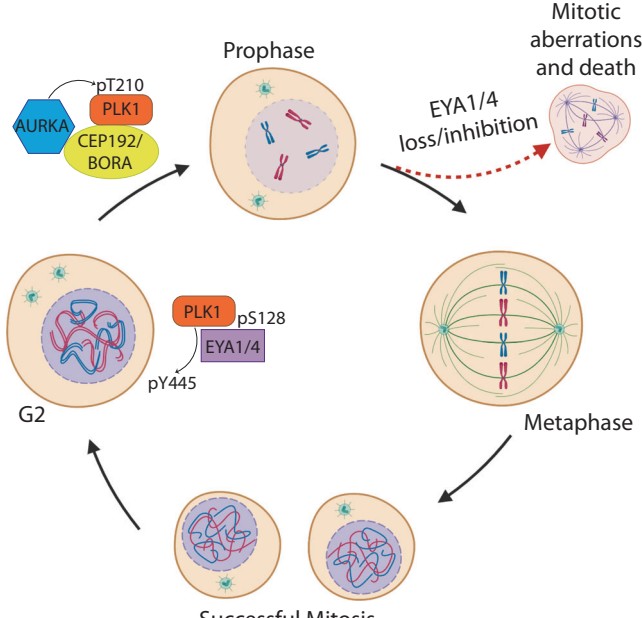

**Fig. 7 | Model depicting dephosphorylation mediated regulation of PLK1 by EYA4/EYA1.** G2, Phosphorylation of the putative PDS on EYA4/EYA1 (S128 on EYA4) allows for interaction with PLK1 and dephosphorylation of pY445 within the PBD. G2→Prophase, Dephosphorylation of pY445 changes the structure of the PLK1 PBD, favouring the interaction between PLK1 and PLK1 activation complex members, and permitting greater levels of pT210 phosphorylation by AURKA. Prophase, Active PLK1 supports centrosome maturation and separation in preparation for mitotic spindle formation. Prophase→Successful Mitosis, Accurate and timely completion of mitosis. Alternatively, EYA4/EYA1 loss or EYA inhibition causes mitotic aberrations, increased mitotic duration, and mitotic cell death. Created with biorender.com.

Mechanistically, dephosphorylation of pY445 promotes PLK1 activation in mitosis by supporting the interaction between PLK1 and PLK1-activation complexes. Phosphorylation of Y445 is predicted to modestly decrease the interaction of the PLK1 PBD with a model peptide, as well as dramatically decrease the flexibility within the connecting loop of the PLK1 PBD structure. We expect that the loss of connecting loop flexibility is particularly important for in-vivo interactions. While our simulation results support this conclusion, the MM/GBSA method is predictive, and follow-up empirical studies will be needed to confirm the role of connecting loop flexibility in PBD functionality[60].

We have shown that phosphorylation of pY445 reduces the ability of the PBD to interact with PDS-containing partners. Therefore, we propose that in conditions in which the EYAs are absent and pY445 remains phosphorylated, PDS-containing proteins would require sufficient proximal accumulation to favour interaction with PLK1. When the EYAs are themselves proximally accumulated to PLK1, as is the case for EYA4 at G2 centrosomes, interaction occurs through the putative EYA PDS, and dephosphorylation of pY445 in PLK1 occurs. This makes the PBD more favourable for subsequent interactions, including with AURKA-containing PLK1 activation complexes, thereby enhancing PLK1 activation in the lead-up to mitotic entry.

Overall, we have characterized a new EYA-PLK1 signalling pathway with important implications for the structural regulation of PLK1, centrosome biology, and mitosis. Inactivation of the EYA-PLK1 signalling pathway potently induces mitotic cell death. This work highlights a mechanism by which EYA phosphatase inhibitors may illicit cell killing in a therapeutic context, and justifies testing of EYA phosphatase inhibitors in preclinical tumour models.

## Methods

### Vectors
Expression constructs for PLK1, EYA4, EYA1, and EYA3 were purchased in pCMV6-Myc-DDK backbones from Origene Technologies. EYA4 and PLK1 mutants were generated using gBlocks™ Gene Fragments and restriction cloning. Wild-type PLK1 and PLK1 mutants were also cloned into pLenti-C-Myc-DDK-IRES-Puro backbones obtained from Origene Technologies.

### Cell culture, transfection, and gene knockdown
HeLa, HEK293T, SKNAS, and SKNFI cells were maintained in DMEM with 10% FCS in a sterile humidified incubator at 37 °C with 10% $CO_2$ and 20% $O_2$. All cell lines were mycoplasma free and verified by STR profiling through CellBank Australia. Cells were passaged and harvested with trypsin. In conditions of nocodazole arrest, cells were harvested by mitotic shakeoff. Cells were transfected at approximately 50% confluency with plasmid DNA using FuGENE® 6 or FuGENE® HD at a 3:1 ratio of FuGENE to DNA, according to manufacturer recommendations (Promega). For transient transfections plasmid DNA expression was allowed to proceed for 48 h prior to downstream analysis. Gene knockdown was performed using Lipofectamine™ RNAiMAX according to manufacturer instructions (Thermo Fisher Scientific). Media containing RNAiMax and siRNA was changed, or cells were split into fresh media, 24 h post transfection. Unless otherwise stated, downstream analysis occurred 72 h after initial siRNA transfection. Validation of gene knockdown was performed routinely by western blotting and concurrently in western blotting experiments in one or more biological replicates when there was sufficient sample volume.

siRNAs used in the manuscript included the following: EYA1: Thermo Fisher Scientific (s534823), EYA4: Thermo Fisher Scientific (s4795), Control siRNA: Thermo Fisher Scientific (AM4613), PLK1 3'UTR: Thermo Fisher Scientific (Custom Sequence, 5'-CCCACCA-TATGAATTGTACAGAATA-3')

### Lentiviral overexpression and rescue experiments
Lentivirus from pLenti-C-Myc-DDK-IRES-Puro constructs was produced by the Vector and Genome Engineering Facility (Children's Medical Research Institute). For rescue experiments involving transient lentiviral overexpression (Fig. 5C–J), cells were first transduced with 1 mL of lentivirus solution per T75 followed by siRNA knockdown as described at 48 h. Cells were imaged 24 h later or harvested for western blotting 48 h later. For benzarone dose response curves cells were transduced with lentiviral expression vectors containing WT PLK1, PLK1 phosphomutants, or with an empty vector, allowed to recover for 24 h, and then subjected to puromycin selection for 7 days prior to drug testing.

### Cell synchronization and drug treatments
For enrichment of cells in mitosis or G2, cells were first synchronized in S-phase for 18 h using 2 mM thymidine. Thymidine containing media was then removed and cells were washed 4 times with fresh media. For arrest in mitosis, nocodazole (0.1 ug/ml) was added to cells for 12 h prior to harvest. For arrest in G2, RO-3306 was added to media 4 h after release from thymidine at a concentration of 10 μM and cells were incubated for 6 h prior to harvest. RO-3306 arrested cells were released into mitosis by removal of RO-3306 containing media and 3 washes with fresh media. Benzarone was diluted in DMSO and added to cells at concentrations as listed in the text.

For benzarone dose-response curves, cells were treated with benzarone or DMSO at the concentrations listed followed by incubation for 72 h. Alamar blue assays were performed as per manufacturer instructions (Invitrogen) and read on a spectrophotometer. Analysis was performed on the difference in absorbance at 570 and 600 nM.

## Immunoprecipitation/Co-immunoprecipitation

Cells were lysed in co-immunoprecipitation buffer (20 mM HEPES-KOH pH 7.9, 200 mM NaCl, 10% (v/v) glycerol, 0.1% Triton X-100, 1 mM DTT) supplemented with cOmplete Mini EDTA-free protease inhibitor cocktail and PhoshoStop phosphatase inhibitor tabs (Roche) at 4 °C. Lysates were cleared by centrifugation at 16,000 x g for 25 min at 4 °C. Simultaneously, 4 or 5 ug of appropriate antibodies or IgG control proteins were bound to protein G Dynabeads™ as per manufacturer instructions (Life Technologies). Equal portions of cleared lysates were added to antibody or IgG bound Dynabeads™ and incubated for 3 h or overnight at 4 °C. Beads were washed with co-immunoprecipitation buffer at 4 °C 4x for co-immunoprecipitation and 6x for immunoprecipitation, followed by elution with a 2:1 mixture of 50 mM Glycine, pH 2.8 and 1x NuPAGE™ LDS sample buffer (Life Technologies) at 70 °C for 10 min.

## BioID pulldowns

BioID-Myc or BioID-Myc-EYA4 plasmids were transfected into T150 flasks of 293Ts at 50% confluence in quadruplicate. After 24 h, media was removed and replaced with media containing 50 μM biotin. After 24 h incubation with biotin, cells were trypsinised, pelleted and resuspended in 500 μL GdmCL lysis buffer (6 M GdmCL, 100 mM Tris pH8.5, 10 mM TCEP, 40 mM Iodoacetamide). Lysates were heated at 95 °C for 5 min and then cooled on ice for 15 min. Samples were sonicated using a Bioruptor ® sonicator for 5 duty cycles for 30 s on and 30 s off at 4 °C. Lysates were then heated again at 95 °C for 5 min and then cooled on ice for 15 min, followed by a 30 min RT incubation in the dark to allow for iodoacetamide to react with Cys residues. Lysates were then diluted 1:1 with milliQ water, and protein was precipitated overnight at −20 °C using 4 volumes of acetone. The following day the proteins were pelleted by centrifugation for 5 min at 1500 RPM, washed once with 80% acetone, and resuspended by sonication as before. Proteins were again pelleted by centrifugation, acetone aspirated and left to dry for 15 min. Dried pellets were resuspended by sonication as before in resuspension buffer (2 M urea, 50 mM Sodium Pyrophosphate pH 8, 50 mM Ammonium Bicarbonate, 0.2% SDS). Protein concentration was determined by BCA assay. Next, 2 mg of each lysate was used in overnight immunoprecipitation reactions with 20 mg of streptavidin-coated dynabeads. The following day, the beads were washed for 3 × 5 min with rotation in resuspension buffer. Next, 10% of the beads were transferred to a new tube and subject to elution of biotinylated proteins using 1X LDS at 80 °C for 10 min for downstream western blot analysis. The remaining 90% of the beads were washed 3x with 100 mM Ammonium Bicarbonate and then resuspended in 100 μl of fresh 100 mM Ammonium Bicarbonate with 8% Acetonitrile (ACN) and 5 μg trypsin. On bead trypsin digestion was carried out overnight at 37 °C shaking at 2000 RPM.

## C18 desalting of tryptic peptides

C18 stagetips were made using a syringe punch to pack to 2 layers of C18 filter into a low-retention p200 tip. C18 stagetips were activated by centrifuging through 100 μl of 100% acetonitrile and equilibrated with 0.1% formic acid (FA) and 0.1% trifluoroacetic acid (TFA). Tryptic peptides were prepared for desalting by the addition of 30 uL of 3.2 M KCL, 11 ul 150 mM KH$_2$PO$_4$ and 19 ul of 100% TFA followed by centrifugation at max speed for 15 min and re-tubing. Peptides were partially dried in a speedyvac for 30 min and then added to pre-equilibrated C18 stagetips and centrifuged at 1500 RPM. C18 bound peptides were washed twice with 100 μl of 0.1% FA and 0.1% TFA. Peptides were eluted with 40% ACN in 0.1% FA and fully dried in a speedyvac. Finally, samples were resuspended in 6 μl of mass spec buffer (1% FA, 2% ACN).

## LC-MS/MS and analysis for BioID and pY IPMS

Peptides were separated using a Dionex Ultimate 3000 UHPCL system with a nanoflow silica column packed with 1.9 μM C18 beads using a 195 min gradient. Mass spectrometry was performed using a Thermo Q Exactive instrument. The MS was operated in data-dependent mode with an MS1 resolution of 35,000, automatic gain control target of 3e6, and a maximum injection time of 20 ms. The MS2 scans were run as a top 20 method with a resolution of 17,500, an automatic gain control target of 1e5, a maximum injection time of 25 ms, a loop count of 20, an isolation window of 1.4 m/z and a normalized collision energy of 25.

Raw MS spectra data were matched, and label-free quantification performed using MaxQuant (v1). Default settings were used with the exception of LFQ min count, which was set to 1, matching from and to was enabled as well as matching between runs. Additionally, the match time was set to 1.5 min. For matching, the following FASTA files were used: UP000005640_9606 (223,281 entries), UP000005640_9606_additional.fasta (546,697 entries). Raw excel files were exported to Perseus (v2) where data was log2 transformed, potential contaminants removed, and imputation performed. P-values were generated in Perseus using a Student's T-test with a false discovery rate of 0.01. The presence of EYA4 and the established EYA4 interactors SIX1 and SIX2 among the high-confidence EYA4 interactors in the BioID dataset further validated the approach.

For the analysis of BioID interactors with localization to mitotic structures pertaining to Fig. 1B we reduced the stringency for inclusion to include medium confidence EYA4 interactors (Log2 fold difference >1). Subcellular localization data was taken from uniprot as well as primary literature searches for each protein in combination with the search term "mitosis."

## Identification and quantification of PLK tyrosine phosphorylated sites by mass spec

**Pilot experiments.** Following depletion of EYA4, or treatment with a control siRNA, Myc-tagged PLK1 was overexpressed in 293 T cells. 48 h later, cells were treated with 10 μM benzarone for 2 h. Cells were pelleted, and PLK1 was immunopurified and eluted as described in the immunoprecipitation/coimmunoprecipitation section. Immunopurified PLK1 was resolved on 4–12% Bis-Tris precast mini gels (Life Technologies). Following electrophoresis, gels were subjected to colloidal Coomassie staining according to the Anderson method[61]. Bands corresponding to PLK1-Myc were excised, destained, reduced, alkylated, and digested in-gel with trypsin, following a previously described protocol[62]. Peptides were fully dried in a speedyvac and resuspended in 6 μl of mass spec buffer.

Peptides were separated using a Dionex Ultimate 3000 UHPCL system with a nanoflow silica column packed with 1.9 μM C18 beads using an 85 min gradient. The MS was operated in data-dependent mode with an MS1 resolution of 35,000, automatic gain control target of 3e6, and a maximum injection time of 20 ms. The MS2 scans were run as a top 10 method with a resolution of 35,000, an automatic gain control target of 1e5, a maximum injection time of 120 ms, a loop count of 10, an isolation window of 1.4 m/z and a normalized collision energy of 27. Peptides were searched in proteome discoverer.

These experiments identified phosphotyrosine-containing peptides corresponding to pY445, and a peptide with equal likelihood of being phosphorylated at pY425 or pY421.

**PRM experiments.** Myc-tagged PLK1 was overexpressed in triplicate or quadruplicate for each treatment in 293 T cells. Treatments included siRNA against EYA4 or a control sequence, or incubation with benzarone (10 μM) or DMSO for 2 h prior to cell harvesting. Cells were pelleted, and PLK1 was immunopurified and eluted as described in the previous section. Peptides corresponding to PLK1 were prepared as described in the previous section.

Peptides were separated using a Dionex Ultimate 3000 UHPCL system with a nanoflow silica column packed with 1.9 μM C18 beads using a 130 min gradient. Mass spectrometry was performed using a

Thermo Q Exactive instrument operating in parallel reaction monitoring (PRM) mode using an inclusion list populated with phospho or unmodified mass and charge peptide data generated from pilot experiments (Supplementary Data 2). Targeted peptides included pY445, a peptide corresponding to both pY425 and pY421, as well as mass and charge data that corresponded to pT210 and which we reasoned might allow for detection of previously observed pY217 peptides[59]. MS2 scans were performed with a resolution of 17,500, an automatic gain control target of 5e4, a maximum injection time of 50 ms, an isolation window of 1.6 m/z and a normalized collision energy of 30.

Skyline (v19) was used to extract chromatogram data and identify peaks from the PRM scans. The sum of product ion peak areas for individual peptides was quantified in accordance with the Skyline PRM tutorial. The number of fragment ions selected for analysis in Skyline was 6 and we have included quantitative data only when 6 fragment ions could be reliably called in 3 reps from each treatment group. Additionally, data were only included for peptides that had dotP scores ≥ 0.8 across all reps. The ratio of phosphopeptide peak intensities was taken relative to corresponding unmodified peptides.

### In-vitro phosphatase assay
The catalytic domain of human EYA4 (residues 367–639) was subcloned as a poly-His fusion construct with a TVMV cleavage site in the vector pDEST-527. Fusion protein was purified by Ni-NTA chromatography followed by ion exchange (Fast-Q) and size-exclusion chromatography over a Superdex-75 column. Phosphopeptides were synthesized by Lifetein. pY445-scramble was designed using a web-based random number generator to randomize the amino acid positions of pY445. Peptide assays were conducted in 20 mM MES pH 6, 150 mM NaCl, 2 mM $MgCl_2$, using a range of peptide concentrations (0–400 μM) and the Biomol assay. Data was analysed using non-linear regression in GraphPad PRISM assuming Michaelis-Menten kinetics.

### Immunoblotting
Western blot lysates were produced in RIPA buffer (Pierce #89900) supplemented with cOmplete Mini EDTA-free protease inhibitor cocktail and PhoshoStop phosphatase inhibitor tabs (Roche) and 1 mM DTT at 4 °C. Lysates were cleared by centrifugation at 16,000 x g for 25 min at 4 °C. Proteins were separated using 3–8% or 7% Tris-Acetate, or 4–12% Bis-Tris precast mini gels (Life Technologies) and transferred to PVDF membranes at 70 V for 2 h or overnight (Immobilon P, Millipore). Ponceau S staining was used to verify even transfer (Sigma Aldrich). Membranes were optionally cut to detect multiple proteins. Blocking and antibody incubations were then performed with either 5% non-fat milk or bovine serum albumin (fraction V) in PBST or TBST. Bands were visualized with HRP-conjugated secondary antibodies (DAKO) followed by the application of a chemiluminescent reagent (Thermo Scientific). Stripping with Restore™ PLUS stripping buffer (Thermo Scientific) and reprobing was performed in some cases when the subsequent primary was of a different species; however, blots were never stripped more than once. Densitometry was performed in ImageJ on unaltered images.

### Immunofluorescence and Click-IT EdU staining
Cells were grown on coverslips pre-treated with Alcian blue stain to promote adherence (Sigma). For experiments involving EdU incorporation, EdU was added to the media for 1 hr prior to fixation at a final concentration of 10 μM. Cells were washed with PBS and then fixed using freshly prepared 4% paraformaldehyde (Sigma) for 10 min at RT followed by 2X PBS washes and then permeabilization with KCM buffer (120 mM KCl, 20 mM NaCl, 10 mM Tris pH 7.5, 0.1% Triton), or 0.2% Triton in PBS. If required, Click-It EdU staining was performed as per the manufacturer's recommendations (Thermo Scientific). Next, blocking was performed using antibody-dilution buffer (20 mM

Tris–HCl, pH 7.5, 2% (w/v) BSA, 0.2% (v/v) fish gelatin, 150 mM NaCl, 0.1% (v/v) Triton X-100 and 0.1% (w/v) sodium azide) for 1 h at RT. Cells were incubated with primary antibodies for 1 hr at RT or overnight at 4 °C followed by 3 × 10 min washes in PBS. Cells were then incubated with Alexa Fluor conjugated secondary antibodies (Thermo Scientific) at a 1:500 or 1:750 dilution for 1 h at RT. Cells were again washed for 3 × 10 min in PBS followed by staining in DAPI solution for 20 min (Sigma). Coverslips were mounted on slides in ProLong™ Gold antifade. Images were acquired with a Zeiss Axio Imager microscope.

### Antibodies and antibody validation
The following primary antibodies were used in the manuscript: The following antibodies were used in the manuscript: PLK F-8 SC17783 Santa Cruz (1:400), Phospho tyrosine 8954 Cell Signaling (1:1000), EYA4 ab93865 Abcam (1:1000), Actin a2066 Sigma (1:1000), Myc 9B11 Cell Signaling (1:1000), Myc 2278S Cell Signaling (1:1000), BUB1 B-3 SC365685 Santa Cruz (1:1000), p-S10 H3 3377 Cell Signaling (1:1000), p-S10 H3 9706 Cell Signaling (1:1000), H3 9715 Cell Signaling (1:1000), p-T210 PLK1 Ab39068 Abcam (1:100 IF, 1:500 WB), p-T210 PLK1 9062 Cell Signaling (1:100), Pericentrin Ab28144 Abcam (1:100), Pericentrin Ab448 Abcam (1:1000), Alpha Tubulin Ab7291 (1:1000) Abcam, Bora 12109 Cell Signaling (1:1000), Cep192 A302324A Bethyl (1:1000), Aurka 12100 Cell Signaling (1:1000), PARP 9542 Cell Signaling (1:1000), Vinculin V9131 Sigma (1:1000), EYA1 22658-1-AP Protein Tech (1:1000), p-S46 TCTP 5251 Cell Signaling (1:200), p-S133 Cyclin B1 4133 Cell Signaling (1:200), p-S198 cdc25C 9529 Cell Signaling (1:200).

The following secondary antibodies were used in the manuscript: Anti-Mouse AF488 A21202 Thermo Scientific 1:500-1:750, Anti-Rabbit AF488 A32790 Thermo Scientific 1:500-1:750, Anti-Rabbit AF568 A10042 Thermo Scientific 1:500-1:750, Anti-Mouse AF594 A21203 Thermo Scientific 1:500-1:750, Anti-Mouse HRP P0447 Agilent (1:2000), Anti-Rabbit HRP P0448 Agilent (1:2000).

To validate antibodies for the detection of pT210 by immunofluorescence (39069 Abcam, 9062 Cell Signaling), we evaluated the staining intensity following depletion of PLK1 alongside simultaneous evaluation of total PLK1 (17783 Santa Cruz). ab39069 produced primarily nuclear staining and was thus evaluated within nuclei, whereas cs9062 was found in both the nuclei and in the cytoplasm and was evaluated in both locations. As reported, ab39069 was only partially sensitive to PLK1 depletion ($p \leq 0.01$, Supplementary Fig. 2C).[63] Both the nuclear and cytoplasmic signals detected with 9062 were reduced by similar levels as for total PLK1 (Supplementary Fig. 2D, E $p \leq 0.0001$).

### Imaging and image analysis
Fixed cell microscopy images were acquired with a Zeiss Axio Imager microscope. For automated image analysis images were converted from (.CZI) to (.TIFF) and imported into Cellprofiler (v2, v4)[64]. Using custom image analysis pipelines we employed intensity-based thresholding strategies to mask individual objects (nuclei, centrosomes) and intensity-based measurements were made within these masks using the MeasureObjectIntensity module. Cytoplasmic area was approximated by expanding nuclear objects by 25 pixels in every direction. For analysis of centrosome number and intensity in prophase cells, images were imported into ImageJ. Prophase was staged by nuclear morphology and the positivity for H3 S10 phosphorylation and centrosomes were identified using the manual selection tool and intensity determined using the measure function. Mitotic defects were scored manually in image J.

### Live cell imaging and analysis using the Incucyte
For siRNA experiments knockdowns were performed in T75 flasks. Forty-eight hours later cells were reseeded into multiwell plates at 50% confluence and added to the incucyte. Imaging and data collection was

performed 72 h after initial knockdown. Overexpression experiments were performed directly within multiwell plates and imaging and data collection performed 48 h after overexpression. Drug treatments were performed four hours before imaging and data collection. Phase contrast images were taken at 10X at fixed intervals for given experiments.

Mitotic events were scored manually by morphology. The mitotic duration was defined as the elapsed time between mitotic entry and exit. Mitotic entry was defined as the transition of morphologically planar cells to a spherical shape. The mitotic exit was defined as the completion of cytokinesis. Mitotic cell death events were scored based on subjective morphological assessment and were characterized by morphological features including blebbing, cellular fragmentation, and cell bursting.

### Molecular dynamics simulations

**Preparation of structures.** The crystal structure of the polo-box domain (PBD) of polo-like kinase 1 (Plk1) bound to a consensus phosphopeptide (PDB code 1 UMW[32], chains A and E) was used as the initial structure for molecular dynamics (MD) simulations. The N- and C-termini of PLK1 PBD were capped by an acetyl group and N-methyl group, respectively. The protonation states of residues were determined by PDB2PQR.[65] In total, seven PBD systems were set up: apo PBD, PBD complexed with phosphopeptide, apo PBD with Y445 phosphorylated (PBD-pY445), PBD-pY445 complexed with phosphopeptide, apo PBD with Y425 phosphorylated (PBD-pY425), PBD-pY425 complexed with phosphopeptide, and PBD-Y445F complexed with phosphopeptide. Each system was solvated with TIP3P water molecules[66] in a periodic truncated octahedron box such that its walls were at least 10 Å away from the protein, followed by charge neutralization with sodium or chloride ions.

**Molecular dynamics.** Five independent MD simulations with different initial atomic velocities and pseudorandom number generator seeds were carried out on each of the PBD systems. Energy minimisation and MD simulation were performed with the PMEMD module of AMBER 18[67] using the ff14SB[68] force field. Parameters for phosphorylated residues were used as described by ref. 69 A time step of 2 fs was used and the SHAKE algorithm[70] was implemented to constrain all bonds involving hydrogen atoms. The particle mesh Ewald method[71] was used to treat long-range electrostatic interactions under periodic boundary conditions. A cutoff distance of 9 Å was implemented for nonbonded interactions. The non-hydrogen atoms of the protein and peptide were kept fixed with a harmonic positional restraint of 2.0 kcal mol$^{-1}$ Å$^{-2}$ during the minimization and equilibration steps. Energy minimization was carried out using the steepest descent algorithm for 1000 steps, followed by another 1000 steps with the conjugate gradient algorithm. Gradual heating of the systems to 300 K was carried out at constant volume over 50 ps followed by equilibration at a constant pressure of 1 atm for another 50 ps. The restraints were removed for subsequent equilibration (2 ns) and production (300 ns) runs, which were carried out at 300 K and 1 atm, using a Langevin thermostat[72] with a collision frequency of 2 ps$^{-1}$ and a Berendsen barostat[73] with a pressure relaxation time of 2 ps, respectively. The PBD-pY425–phosphopeptide complexes had a longer production time of 500 ns because the standard deviation for the binding free energies were found to be high at the end of 300 ns, indicating non-convergence.

**Binding free energy calculations.** Binding free energies for the PBD–peptide complexes were calculated using the molecular MM/GBSA method[55] implemented in AMBER 18.[67] Two hundred equally-spaced snapshot structures were extracted from the last 100 ns of each of the trajectories, and their molecular mechanical energies calculated with the sander module. The 10 closest water molecules to the phosphothreonine in the phosphopeptide ligand in each snapshot were preserved and considered as part of the receptor for the MM/GBSA calculations. The polar contribution to the solvation-free energy was calculated by the pbsa[74] program using the modified generalized Born (GB) model described by ref. 75, with the solute dielectric constant set to 2 and the exterior dielectric constant set to 80. The nonpolar contribution was estimated from the solvent-accessible surface area using the molsurf[76] program with $\gamma = 0.005$ kcal Å$^{-2}$ and $\beta = 0$. The contribution of entropy was considered to be the same for all complexes and therefore ignored, as the receptors are structurally very similar and the ligand is unchanged.[60]

### Statistical analysis and sample handling

When not otherwise stated in the text *p*-values were generated in GraphPad Prism (v7-10) by two-tailed Students t-tests for comparisons involving two groups, or one-way ANOVAs for three or more groups. Tukey's or Dunnett's multiple comparisons tests were used with ANOVA to generate individual *p*-values when the mean of each group was compared to every other group or when the mean of each group was compared to the control group respectively. Throughout the figures error bars represent standard error of the mean unless otherwise specified. Replicate measurements were made on distinct samples.

### Reporting summary

Further information on research design is available in the Nature Portfolio Reporting Summary linked to this article.

## Data availability

All relevant data supporting the findings of this study are provided in the main figures and/or its supplementary information files. The MD simulation input files, initial and final coordinate files have been deposited in Zenodo with the, https://doi.org/10.5281/zenodo.10223834[77]. The mass spectrometry data have been deposited to the ProteomeXchange Consortium via the PRIDE partner repository with the dataset identifiers PXD036405, PXD036430, and PXD037064. The crystal structure with the PDB identifier 1UMW [https://doi.org/10.2210/pdb1UMW/pdb] was retrieved from the Protein Data Bank (https://www.rcsb.org). Source data are provided with this paper.

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

## Acknowledgements

The authors acknowledge the CMRI ACRF Telomere Analysis Centre supported by the Australian Cancer Research Foundation; the CMRI Vector and Genome Engineering Facility; and the CMRI Biomedical Proteomics Facility. The authors acknowledge support from Luminesce Alliance. The Alliance is comprised of five partners, Sydney Children's Hospitals Network, CMRI, Children's Cancer Institute, University of Sydney, and University of New South Wales. HAP is supported by the National Health and Medical Research Council of Australia (2003250, 1187606, 1162886), and the Medical Research Future Fund (2007488). RSH is supported by NIH funding (CA207068, HL152094). YST is supported by the Bioinformatics Institute, A*STAR.

## Author contributions

CBN and HAP conceived the study. CBN designed the experiments. CGT cloned the BioID plasmids. SR advised on the mass spec experiments and helped with data analysis and troubleshooting. KR and RSH performed the in-vitro phosphatase assays. YST performed the molecular dynamics simulations and wrote or advised the writing on these experiments. CBN performed all other experiments and wrote the manuscript. All authors, including CJA, APS, AH, RL, ED, MH, JIF, and AJC provided intellectual input and advice on the manuscript.

## Competing interests

The authors declare no competing interests.
