## [Peer Review File · Nature Communications]

The Eyes Absent family members EYA4 and EYA1 promote PLK1 activation and successful mitosis through tyrosine dephosphorylationREVIEWER COMMENTS

Reviewer #1 (Remarks to the Author):

In "The Eyes Absent family members EYA4 and EYA1 promote PLK1 activation and successful mitosis through tyrosine dephosphorylation," Nelson et al report compelling evidence that the Polo-like kinase 1 (PLK1) is a direct substrate of the EYA1 and EYA4 phosphatases, and they also identify pY445 of PLK1 as the target's key dephosphorylation site. The authors marshal an impressive range of genetic, proteomic, biochemical, and computational techniques that collectively tell a complete story: during G2, PLK1 associates with and is dephosphorylated at pY445 by EYA4/1 at the centromeres, and dephosphorylation of PLK1 enhances its interaction with PLK1-activation complexes, promoting PLK1 activation and mitotic progression. These findings are of significant biological importance, as EYA phosphatase activity plays oncogenic roles in a range of tumor types. The study is expertly executed and presented and has high potential impact, and I am happy to recommend it for publication in Nature Communications after one minor revision described below.

Figure 4C: In presenting in vitro phosphatase assays, the authors make an arbitrary distinction between "specific" peptides that are dephosphorylated with Michaelis-Menten kinetics" and "non-specific" peptides that are purported to be dephosphorylated with linear kinetics, and the data from the two "different kinds" of peptides are plotted on separate sets of axes. Inspection of the actual data reveal, however, that there is no such distinction. All of the investigated peptides have high K_m values, and, for the "non-specific" peptides that have the highest K_m values, the authors have not gone to high enough substrate concentrations to accurately determine the K_m s. (Also, it should be noted that if the data for pY217 peptide were plotted in the left panel, its data would look exceedingly similar to the "specific" peptides.) The authors should go to higher substrate concentrations, if possible, to determine the kinetic constants for all of the peptides. If solubility issues preclude testing higher concentrations, that should be stated. Regardless, all of the peptide kinetic data should be plotted on the same set of axes, and the revised manuscript should not imply that the "non-specific" peptides cannot be investigated with MM kinetics. Rather, it should simply state that they have higher K_m values.

Reviewer #2 (Remarks to the Author):

Nelson et al. explore the functional consequences of tyrosine phosphorylation on the mitotic kinase Plk1. This is an underexplored area, and new findings would be novel and interesting. Specifically, the authors report that phosphorylation of EYA1 and EYA4 phosphatases at S128 (EYA4) allows interaction with the Polo-box Domain (PBD) of Plk1. In turn, EYA4 dephosphorylates Y445 of Plk1, which facilitates Plk1 activation in cells (as determined by increased T210 phosphorylation), perhaps by enhancing PBD-mediated interaction with activating proteins. Depletion of EYA1/4 causes defects in mitosis that are consistent with loss of Plk1 function. Overall, this is a potentially informative study that reports new findings that may further our understanding of Plk1 activation at the G2 to M transition. However, to have sufficient confidence in the interpretation of the results, there are some experimental issues that should be addressed prior to publication.

Major points

1. Do endogenous Plk1 and EYA1/4 interact? Not essential to show, but would be valuable.
2. None of the experiments characterizing Plk1-EYA1/4 interaction actually demonstrate phospho-dependence or a requirement for the PBD, only that the interaction is altered by S128 mutation. Does Plk1-PBD interact differentially with phosphorylated and dephosphorylated EYA1/4, or with phospho/non-phospho peptides of EYA1/4? Is the interaction abolished by mutation of the PBD

"pincer" residues? This is important given point 11, below.

3. The S128D mutant proteins appear overloaded in the lysates in Figure 1F. Can this experiment be repeated and quantified to support the results (as in other figures)? Also, it is quite surprising that S128D is sufficient to mimic phosphorylation for PBD binding. I am not aware of any precedent for this in the literature for the PBD, but of course I may have missed something. Does structural analysis/molecular dynamic simulation provide any support for this idea for PBD-binding regions?

4. A major concern is the use of the Ab39068 anti-Plk1 pT210 antibody in many figures in the paper. There is good evidence that, by IF, this antibody does not recognize Plk1, though it does recognize an Aurora-kinase dependent epitope on an unidentified protein. Specifically, IF using this antibody does not produce the expected staining of centrosomes in mitosis, and RNAi of Plk1 does not reduce the staining at kinetochores. These findings have been reported by two independent labs as reported in the "Questions/reviews" section on the relevant Abcam webpage, and in one publication (Bruinsma et al. J Cell Sci (2014) 127 (4): 801–811 doi.org/10.1242/jcs.137216). It is possible that this batch of the antibody is different from that characterized previously, and/or that the reactivity seen by western blot does reflect Plk1 (see Bruinsma et al.), but this needs to be validated experimentally. For example, does RNAi of Plk1 eliminate the IF and/or WB signals using this (or an alternative) antibody?

5. Does the phosphorylation of any Plk1 substrate actually change when EYA1/4 and/or Plk1 phosphosite mutations are made? (This would also reduce reliance on the Plk1-T210D antibody to measure Plk1 activity).

6. Uniquely in Figure 2, Figure 2C does not correct the pT210 quantification for Plk1 levels. This does appear to be done in Fig S2C, where an "ns" is buried in the legend which presumably means "not significant". If so, and this means that the change in Plk1 phosphorylation is not significant in this experiment, then this is a very unfortunate approach to reporting results that is misleading and should be rectified.

7. There seems to be little evidence that benzarone is a selective inhibitor of EYA phosphatases. It might have many off-targets. This should be acknowledged more explicitly. For example, in the Discussion, the most likely reason for the difference in RNAi and inhibition phenotypes would seem to be such off-target effects, but this is not mentioned (lines 256-261).

8. Throughout the paper, WBs need molecular weight markers, and the legends should explicitly specify what "n" is. We need to know the number of cells quantified, and in how many independent experiments the observations were made, and we also need to know how "n" was defined for statistical tests in each experiment. All the antibodies used should be specified in the paper.

9. Line 157, the statement that "These data demonstrate that EYA4 and EYA1 function redundantly to promote the accurate completion of mitosis through the dephosphorylation of PLK1" is certainly too strong in my view. Without showing that Plk1, or a mutant thereof, can rescue the defects caused by EYA1/4 loss, we only have correlation, not causation. Does Y445F rescue the effect of EYA1/4 loss?

10. Experiments of the type in Figure 5B are not compelling. Loss of H3S10 phosphorylation in a WB could be caused by either failure to enter mitosis, or a shortening in the duration of mitosis. Any defect in cell cycle progression outside mitosis will cause loss of H3S10 phosphorylation. As Plk1 is implicated in mitotic entry, this is more than a theoretical concern. Live imaging of mitosis, as in Figure 3, would be needed to substantiate this effect. The fact that Plk1 WT does not rescue Plk1 RNAi is also curious. Why is this?

11. A potential problem with the model is that EYA1/4 is proposed to interact with Plk1-pY445 through the PBD of Plk1. However, pY445 is proposed to reduce the affinity of the PBD for its binding partners. How then does EYA1/4 bind to, dephosphorylate, and activate Plk1? Does EYA1/4 have higher affinity for the PBD compared to other binding proteins, even when Y445 is phosphorylated? Is pS128 really a PBD-binding site (see point 2 and 3)?

Minor points

1. Line 40. I would say Cdk1-CycB is the master regulator of mitosis, not Plk1.

2. Line 81/Figure S1F. It would be good to show an alignment of the EYA sequences with the PDS consensus.

3. Separation, not seperation in Figure 1G.

4. Classifying as "1 or 2 centrosomes" in Figure 3 is confusing. The model seems to be that EYA1/4

affects centrosome separation, not duplication. Perhaps "1 or 2 centrosome-containing foci", or similar, would be better.

5. Please specify how mitotic duration and mitotic death were determined in Figure 3.

6. The proteins and text inside the cells in Figure 6 are too small to read. Also, S128 does not have very obvious sequence features of a Cdk site, so saying "possibly" a Cdk site perhaps would be better than "likely".

7. It is somewhat surprising that Caron et al. *Science Signaling* (2016) 9, rs14 doi:10.1126/scisignal.aah3525 is not mentioned as context, but not essential.

Reviewer #3 (Remarks to the Author):

In the presented manuscript, authors report that function of a mitotic protein kinase PLK1 is regulated by tyrosine phosphorylation. Searching for potential substrates of a tyrosine phosphatase EYA1, they performed proximity biotinylation and identified PLK1 and several other centrosomal proteins as interacting partners of EYA1. They observed that knock down of EYA1 increased phosphorylation of PLK1 at tyrosine residues and using mass spectrometry, they identified Y445 as a major modified residue. In addition, depletion of EYA1 increased the level of PLK1 modification at pT210 that is used as proxy of PLK1 activation. Depletion of EYA1 (and its relative EYA4) caused mitotic defects similar as inhibition of PLK1 (including impaired centrosomal maturation and separation). Authors concluded that inactive PLK1 is phosphorylated at Y445 in G2 and EYA1-dependent dephosphorylation leads to activation of PLK1 at mitotic entry. Mechanisms controlling PLK1 activation are incompletely understood and also function of tyrosine phosphorylation in cell cycle is largely unexplored. Mechanism described in this manuscript is novel and surely could be of interest in the field. Most of the presented data is convincing but several additional experiments need to be done to allow the main conclusion that PLK1 acts downstream of EYA1/4. Addressing the issues detailed below would also significantly strengthen the manuscript.

Major points

1. Figure 3 shows that EYA1 and EYA4 depletion causes several mitotic defects that resemble inhibition of PLK1. Authors state that EYA4 and EYA1 function redundantly to promote the accurate completion of mitosis through the dephosphorylation of PLK1. However, this conclusion is not supported by the presented data. For this, authors need to show that the mitotic defects are rescued by expression of a PLK1 mutant that cannot be phosphorylated at Y445 or by constitutively active PLK1-T210D.

2. Based on results in Figure 2, authors conclude that EYA1 and EYA4 promote PLK1 activation. This might be the case but several issues need to be clarified. It is unclear how (and why) they quantified pT210 PLK1 signal at kinetochores (Fig. 2A)? As PLK1 and EYA1 colocalize at centrosomes, they should better quantify pT210 PLK1 signal at centrosomes rather than kinetochores? Figures 2B-D show decreased PLK1 pT210 signal upon depletion or inhibition of EYA1 or after expression of S128A EYA1. However, it seems that there is a reduction of pH3-S10 signal in all these conditions suggesting that observed PLK1 modification at T210 might be caused by decreased proportion of mitotic cells. Authors should evaluate PLK1-pT210 signal in mitotic cells collect by mitotic shake off. Benzarone also reduced the expression level of PLK1 in Fig. 2C and therefore authors should plot pT210/PLK1 ratio rather than pT210 level.

3. The staining of pT210 in the T loop of PLK1 might be influenced by modification of the close by residue Y217 that was reported as target of cAbl by Yang et al., 2017. Therefore, authors should formally demonstrate that PLK1 activity is decreased upon manipulation with EYA1. For instance, they can measure the activity of PLK1 immunopurified from mitotic cells by in vitro kinase assay.

4. Fig. 1G – specificity of the staining with EYA4 antibody should be validated (for example by siRNA)

5. The authors should elaborate more on the kinase responsible for Y445 modification. cAbl has previously been implicated in activation of PLK1 by phosphorylating Y217, Y425 and Y445 (Yang et al., 2017). Therefore, there seems to be a conflict with the model proposed here according to which phosphorylation of Y445 keeps PLK1 inactive during G2.

Minor points

1. EYA depletion increased pTyr signal of PLK1 purified from G2 cells but the signal was reduced later in mitosis (Fig 1D). Does it mean that another phosphatase targets PLK1 in mitosis? Why is no change of pTyr levels of PLK1 observed in control depleted cells?
2. The statement that colocalization between EYA4 and PLK1 was strongest in cells prior to centrosome separation (Figure 1G) needs to be supported by quantification. Is there also some colocalization in the nucleus? Are these confocal or wide field images?
3. Authors should consider reformulation of the statement that EYA1 and EYA4 are functionally redundant. In that case, depletion of individual phosphatase should not yield any phenotype.
4. In Fig. 2D, the EYA4-S128A and Ydef mutants have clearly less cells in mitosis and therefore comparison of S128A and S128D is not informative. Instead, authors should evaluate statistical significance between WT and S128D where the amount of mitotic cells is comparable. Fig S2D shows comparison between WT and S128A in non-arrested cells but this is irrelevant as S128A will likely decrease the proportion of mitotic cells.
5. Authors state that S128A has a dominant negative effect (Fig. 3K). I have hard time to understand how this could work if S128A does not bind PLK1. Instead, I would expect that pYdef mutant should have DN effect, but this is not the case as cells progress normally.
6. Is there significant difference between pYdef mutant and the WT in Fig. 3L?
7. What does Y445 scramble mean? It does not resemble the motif at all, so it should be called control pY peptide but not Y445
8. Figure S5A shows that Y445F mutant is strongly phosphorylated at other tyrosine residues. Does this pY signal respond to depletion of EYA1? How does Y217F mutant perform in this assay?

Reviewer #4 (Remarks to the Author):

In the manuscript, Nelson et al. identified EYA4 as a key player for regulating PLK1 phosphorylation status on Y445, and its crucial role in regulating mitosis. Overall, it is an interesting and well written manuscript. The authors applied molecular dynamics simulations to suggest the potential molecular mechanism related to the effect of the phosphorylation. I have focused my review on those MD simulations. The applied methods and methodology are mainly sufficient and sufficiently described. This being said, the conclusion made from the presented results appear a bit thin, and I have the following concerns:

- 1) The timescale of the conducted simulations is short; thus, it is questionable if there is enough time for the proper system relaxation with the modified (phosphorylated) system to describe its biologically relevant configuration / conformational ensembles. In the manuscript, the authors point out that Y445F mutant was found to be hyperactive. I would find more confidence with the simulations (and their timescale) to describe the biologically relevant events/motions if simulation results of Y445F agree with the other presented results (and reflects correctly to the biological observations). The next points below are also of high importance in providing confidence on the simulation observations.
- 2) In figure 5D, an interaction between R507-pY445 is shown. Please provide information of the observed frequency of this R507-pY445 interaction in the simulations. Which interactions R507 display in the non-phosphorylated system? Furthermore, please demonstrate that the simulations are not biased due to the starting configuration of the pY445 system i.e. there is no impact with the close proximity of the R507-pY445 in the start conformation of the simulations (see the next point related to this). What is the role of the other important residues (close to Y445) in the dynamics of phosphorylated and non-phosphorylated systems e.g. Q452; E460; Y462; F409; D429 (e.g. their interactions)?

3) The structure (1UMW) used in the simulations is not the best quality and there is no electron density data deposited with it to the PDB. It is also not mentioned, which chain of the structure was used in the simulations. There are better quality structures with electron densities available (important especially for R507 and the neighbouring residues), which illustrate also different conformations for these loop residues, especially R507 (e.g. PDB IDs: 3BZI; 3CL5; 3FVH; 3P2Z etc.). Would a simulation with a different starting conformation (structure) result in the same observations?

4) Figure 5E, F: Is this an average RMSF value from all the simulated replicas? Are there differences among replicas? Display the SD with the average to demonstrate that there is a difference between the systems (and confidence on the difference between the systems).

The authors could point out that there is observed disorder in the loop in some of the PLK1 structures (e.g. 2OCQ; 2OJX; 3P2W), which is nicely aligning with their RMSF data.

5) It is rather well understood that the MMGBSA free energy estimates are highly suggestive. The authors should point out this limitation for the applied methodology in the manuscript. Also, there is no experimental values to compare here and their validity cannot be evaluated (see point 1 related to this below).

Other points

1) A biophysical assay (e.g. ITC, SPR) demonstrating the discrepancy of binding affinities between Y445 and pY445 (PBD-substrate) would be a highly valuable addition to the manuscript.

2) Fig 5D & Fig 6 labels in the figure are way too small to be easily seen. Please improve the readability.

3) The authors do not state how the simulation data is made readily accessible for the readers. There are easy options, such as public repositories to upload the data (e.g. Zenodo). The out-dated statement "simulation data is available from the authors upon sufficient request" is not sufficient here.

4) The authors have conducted simulations with 4 replicas, although the general recommendations are a minimum of 5–10 replicas. (Knapp et al. J. Chem. Theory Comput. 2018, 14, 12, 6127–6138)

5) In the methods, it is not well explained how the how the four replica simulations are independent.

6) It is unclear why pY425 systems underwent slightly longer simulations, and therefore a different "time" is compared in MMGBSA with it to the other systems.

7) Is there a specific reason that the authors were not using ff19SB force field?

8) Line 223: "The binding free energy of the interaction...". Write more clearly which interaction is referred here.

Reviewer #5 (Remarks to the Author):

Nelson et al. report a multi-faceted study that describes a novel signaling pathway in which EYA4 and EYA1 regulate PLK1 through the dephosphorylation of pY445 during G2. Starting with proximity proteomics data to interrogate the EYA4 interactome, they identify PLK1 among other centrosome protein interactors, and proceed to biochemically characterize its phosphatase activity on Y445 and how this regulates mitotic progression via PLK1 activation. The authors note that this is the first report of its PLK1 Y445 dephosphorylation and functional significance. This appears to be a well-designed

study with multiple lines of supporting evidence. I was asked to review this manuscript to provide perspective on the proteomics data, which is largely sound but requires addition of minor details as noted below.

1.) The authors need to provide proximity proteomics data as a spreadsheet (e.g., Excel file, .csv, .txt) in addition to the pdf format they currently have. Data difficult to examine and evaluate in the pdf format and pdfs are more challenging for informatics pipelines that may be used later to further explore this data.

2.) In the Methods, can the authors please include the number of entries present in the UP000005640_9606 FASTA file? Does it include all the isoforms of the human proteome? Are there any isoforms of EYA4, EYA1, or PLK1 present in the database that would complicate data analysis?

3.) I may have missed it, but how many biological replicates were performed for the proximity proteomics experiment? I see replicates noted for the pTyr site-specific PRM experiments, but I did not see a replicate number that was used for statistics in the first LC-MS/MS experiment.

4.) Why does Supp Fig. 1e only use two replicates for data analysis when the Methods report that experiments were done in triplicate or quadruplicate? Or maybe I am misreading which replicates were used for pTyr site analysis versus pY IP-MS. Regardless, a clearer denotation of experiment design would be helpful. Also, PLK1 fold change is on the edge of significance in the volcano plot from the pY IP-MS data and would not be considered significant using the requirements they established for the proximity proteomics ($FC > 2$). The authors clearly have follow up data to support their conclusions, so this does not change the outcome of the manuscript... but knowing the authors interpretation of a PLK1 pTyr barely making their significance threshold with not much fold change would be a useful addition.

5.) If performing a gene set enrichment on subcellular location terms from all medium confidence and greater EYA4 interactors, is the centrosome/spindle among the top hits, or do EYA4 interactors share multiple subcellular locations? The authors logic on targeting centrosome or spindle proteins for the interactors is sound, but I am curious if this subcellular location is a dominant feature for EYA4 in these experiments. This would lend evidence to a "professional role" of EYA proteins in the centrosome, or it could indicate that there are several other subcellular locations that could be explored in subsequent work.

6.) Can the authors include the methods used for the pilot experiment performed to get the list of (pTyr) peptide targets for PRM? How confident are they that the phosphosite is Y instead of S or T in those peptides? This seems like a key point to address to make sure it is the pTyr site related to EYA phosphatase activity. Annotated spectra showing clear pTyr modification and no S or T mods would be highly conclusive for this.

7.) It appears the Skyline file used for analysis is not included in the PRIDE submissions that have the raw files, but it is possible I missed it. Can the authors include that file or verify that it is currently there? Also, I know they mentioned in the Methods that they followed guidelines presented in the Skyline tutorial, but if they could state some of these metrics (dotp scores, number of fragments used, etc), that would be very helpful to evaluate their strategy. Similarly, results files from the other LC-MS/MS experiments are missing in the PRIDE submissions, but they could be uploaded as supplemental spreadsheets as mentioned above if the authors do not prefer to add the files to the PRIDE submission.

8.) Labels on the lanes in the WBs in Fig.1 could be better described. For example, does the lane (column) labeled "EYA4" mean EYA4 depletion? I assume so based on the text description and my interpretation, but having a description in the legend or a clearer label would help interpretation.

9.) Line 140: "Total spindle defects increased modestly following EYA1 and EYA4 depletion" makes it sound like the combined condition, when really it is the individual conditions. Using "or" instead of "and", or using "individual" in the same fashion that "combined" is used in the next sentence, could solve this confusion.

RESPONSE TO REVIEWERS' COMMENTS

Reviewer #1 (Remarks to the Author):

In "The Eyes Absent family members EYA4 and EYA1 promote PLK1 activation and successful mitosis through tyrosine dephosphorylation," Nelson et al report compelling evidence that the Polo-like kinase 1 (PLK1) is a direct substrate of the EYA1 and EYA4 phosphatases, and they also identify pY445 of PLK1 as the target's key dephosphorylation site. The authors marshal an impressive range of genetic, proteomic, biochemical, and computational techniques that collectively tell a complete story: during G2, PLK1 associates with and is dephosphorylated at pY445 by EYA4/1 at the centromeres, and dephosphorylation of PLK1 enhances its interaction with PLK1-activation complexes, promoting PLK1 activation and mitotic progression. These findings are of significant biological importance, as EYA phosphatase activity plays oncogenic roles in a range of tumor types. The study is expertly executed and presented and has high potential impact, and I am happy to recommend it for publication in Nature Communications after one minor revision described below.

1. Figure 4C: In presenting *in vitro* phosphatase assays, the authors make an arbitrary distinction between "specific" peptides that are dephosphorylated with Michaelis-Menten kinetics" and "non-specific" peptides that are purported to be dephosphorylated with linear kinetics, and the data from the two "different kinds" of peptides are plotted on separate sets of axes. Inspection of the actual data reveal, however, that there is no such distinction.

All of the investigated peptides have high K_m values, and, for the "non-specific" peptides that have the highest K_m values, the authors have not gone to high enough substrate concentrations to accurately determine the K_m s. (Also, it should be noted that if the data for pY217 peptide were plotted in the left panel, its data would look exceedingly similar to the "specific" peptides.) The authors should go to higher substrate concentrations, if possible, to determine the kinetic constants for all of the peptides. If solubility issues preclude testing higher concentrations, that should be stated. Regardless, all of the peptide kinetic data should be plotted on the same set of axes, and the revised manuscript should not imply that the "non-specific" peptides cannot be investigated with MM kinetics. Rather, it should simply state that they have higher K_m values.

Thank you for this detailed comment. The peptides had solubility issues over 400 μ M and could not be tested at higher concentrations. This has been stated in the revised manuscript. As requested, we plotted all the peptide kinetic data together on one graph to facilitate direct comparisons between the peptides (see below). However, we feel that there remains a clear difference between the two groups of peptides across the substrate concentrations tested, making the separate graphs easier to interpret. We have therefore included the merged graph in Figure S4 and retained the separate graphs in Figure 4. This is also reflective of our decision to pursue pY445/pY425 in part based on these data. For further clarity, we have changed the figure legend to reflect that dephosphorylation was either "fast" or "slow" as opposed to Michaelis-Menten and linear, we have amended the order of listed peptides to match the order on the graph, and we have included the K_m values in the figure legend.

As the reviewer states, all the investigated peptides have high K_m values, but it is noteworthy that the best-established EYA substrate (H2AX pY142) has a similar (slightly higher) K_m to pY425 and pY445, suggesting that PLK1 is dephosphorylated at similar kinetics to other EYA substrates *in vitro*.

Reviewer #2 (Remarks to the Author):

Nelson et al. explore the functional consequences of tyrosine phosphorylation on the mitotic kinase Plk1. This is an underexplored area, and new findings would be novel and interesting. Specifically, the authors report that phosphorylation of EYA1 and EYA4 phosphatases at S128 (EYA4) allows interaction with the Polo-box Domain (PBD) of Plk1. In turn, EYA4 dephosphorylates Y445 of Plk1, which facilitates Plk1 activation in cells (as determined by increased T210 phosphorylation), perhaps by enhancing PBD-mediated interaction with activating proteins. Depletion of EYA1/4 causes defects in mitosis that are consistent with loss of Plk1 function. Overall, this is a potentially informative study that reports new findings that may further our understanding of Plk1 activation at the G2 to M transition. However, to have sufficient confidence in the interpretation of the results, there are some experimental issues that should be addressed prior to publication.

Major points

1. Do endogenous Plk1 and EYA1/4 interact? Not essential to show, but would be valuable.

We were unable to demonstrate a direct interaction between endogenous Plk1 and EYA1/4. We attribute this to EYA4 being expressed at low levels and/or the antibody that detects it having low affinity, making co-immunoprecipitations involving endogenous EYA4 challenging. Nevertheless, we provide several other indications of a bona-fide interaction between PLK1 and EYA4, including co-immunoprecipitation with overexpressed EYA4, colocalization via immunofluorescence, proximity labelling of PLK1 via EYA4 bio-ID constructs, as well as compelling evidence that EYA4 alters PLK1 tyrosine phosphorylation in cells.

2. None of the experiments characterizing Plk1-EYA1/4 interaction actually demonstrate phospho-dependence or a requirement for the PBD, only that the interaction is altered by S128 mutation. Does Plk1-PBD interact differentially with phosphorylated and dephosphorylated EYA1/4, or with phospho/non-phospho peptides of EYA1/4? Is the interaction abolished by mutation of the PBD "pincer" residues? This is important given point 11, below.

Thank you for this comment. We have amended the discussion to clarify that our data using the S128 phospho-mutants are indicative of a PDS-mediated interaction, and that the interaction between PLK1 and EYAs is likely to involve the PBD.

3. The S128D mutant proteins appear overloaded in the lysates in Figure 1F. Can this experiment be repeated and quantified to support the results (as in other figures)? Also, it is quite surprising that S128D is sufficient to mimic phosphorylation for PBD binding. I am not aware of any precedent for

this in the literature for the PBD, but of course I may have missed something. Does structural analysis/molecular dynamic simulation provide any support for this idea for PBD-binding regions?

Densitometry of western blots has been provided throughout the manuscript. Figure 1F has been repeated and the differences in EYA4 mutant interactions with PLK1 are reliable. We do not typically consider co-immunoprecipitations to be quantifiable, but we have now performed densitometry and included the quantifications in Figure 1F. Quantification indicates that, despite loading differences, the G2 interaction between PLK1 and S128A is reduced, whereas the interaction with S128D is increased. Persistence of the S128D interaction into mitosis is moderate, and this has been clarified in the text.

There is precedence for phosphomimetic mutants mimicking polo-docking site phosphorylation, as described for similar mutations in the PLK1 interactors, AMPK α 2, INCENP, Mis18a, PTP1B, MYC, and CLASP2 (Lu, Jianlin, et al., 2021; Papini, Diana, et al., 2019; Lee, Minkyung, et al., 2018; O'Donovan, D. S., et al., 2013; Padmanabhan, Achuth, et al., 2013; Maia, Ana, et al., 2012).

4. A major concern is the use of the Ab39068 anti-Plk1 pT210 antibody in many figures in the paper. There is good evidence that, by IF, this antibody does not recognize Plk1, though it does recognize an Aurora-kinase dependent epitope on an unidentified protein. Specifically, IF using this antibody does not produce the expected staining of centrosomes in mitosis, and RNAi of Plk1 does not reduce the staining at kinetochores. These findings have been reported by two independent labs as reported in the "Questions/reviews" section on the relevant Abcam webpage, and in one publication (Bruinsma et al. J Cell Sci (2014) 127 (4): 801–811 doi.org/10.1242/jcs.137216). It is possible that this batch of the antibody is different from that characterized previously, and/or that the reactivity seen by western blot does reflect Plk1 (see Bruinsma et al.), but this needs to be validated experimentally. For example, does RNAi of Plk1 eliminate the IF and/or WB signals using this (or an alternative) antibody?

Thank you for this comment. We were unaware of potential issues with this antibody. We have now verified the western blot data in Figure 5a using an alternative anti-Plk1 pT210 antibody (CAT# 9062 Cell Signalling), see below.

To validate the use of antibody Ab39068 in immunofluorescence (IF), we knocked down PLK1 with siRNA and examined total PLK1 staining in the nuclei of mitotic cells, as well as pT210 staining by Ab39058. Both total PLK1 and pT210 staining were significantly reduced compared to control siRNA, however the magnitude of the effect was not as great for pT210. This may indicate some non-specific staining with Ab39058.

We therefore tested the second antibody (CAT# 9062 Cell Signalling) in IF experiments. Similarly, 9062 did not produce robust staining at centrosomes in unsynchronised HeLa cells but did stain both in the cytoplasm and in the nucleus (Figure 2A). Both the nuclear and cytoplasmic signals detected with 9062 were reduced following siRNA depletion of PLK1 by similar levels as for total PLK1, indicative of improved selectivity of antibody 9062 to pT210. For this reason, we repeated the experiment presented in Figure 2A in HeLa cells using 9062. The results were comparable to those obtained with Ab39058, providing additional confidence that mitotic pT210 levels are reduced by depletion of the EYAs. Additionally, because 9062 is a rabbit monoclonal antibody, we were able to use it in combination with our PLK1 antibody (mouse) to obtain per nuclei pT210/PLK1 ratios, further supporting the effects on PLK1 pT210 levels occurring at a per-molecule level.

We have now replaced the data in Figure 2 with data generated with antibody 9062, and moved the data generated with antibody Ab39058 to Figure S2A. Thorough antibody validation is now provided in Figure S2C-E.

5. Does the phosphorylation of any Plk1 substrate actually change when EYA1/4 and/or Plk1 phosphosite mutations are made? (This would also reduce reliance on the Plk1-T210D antibody to measure Plk1 activity).

We have now performed IF for three PLK1 substrates (pS46 TCTP, pS133 CycB, pS198 CDC25C) in unsynchronized mitotic cells following depletion of EYA4. The cellular intensities of all three substrates decreased following EYA4 depletion, consistent with our conclusions. These data have been included in Figure 2D-G of the revised manuscript.

6. Uniquely in Figure 2, Figure 2C does not correct the pT210 quantification for Plk1 levels. This does appear to be done in Fig S2C, where an “ns” is buried in the legend which presumably means “not significant”. If so, and this means that the change in Plk1 phosphorylation is not significant in this experiment, then this is a very unfortunate approach to reporting results that is misleading and should be rectified.

Please accept our apology for the confusion. We did not mean to be misleading. We have moved the pT210/PLK1 ratio data to the primary figure (now Figure 2J) and amended the accompanying text to discuss this observation.

7. There seems to be little evidence that benzarone is a selective inhibitor of EYA phosphatases. It might have many off-targets. This should be acknowledged more explicitly. For example, in the Discussion, the most likely reason for the difference in RNAi and inhibition phenotypes would seem to be such off-target effects, but this is not mentioned (lines 256-261).

Benzarone has been shown on multiple occasions to inhibit EYAs in-vitro. Additionally, there are several papers that use benzarone to produce effects that mirror genetic experiments involving the EYAs. However, off target effects are always a possibility, and inhibition of the EYAs including EYA2 and EYA3, could also contribute to our observations. We have acknowledged these possibilities explicitly in the revised manuscript, cited the major papers relevant to benzarone, and included data in Figure 5 showing that expression of the PLK1 Y445F mutant provides some resistance to viability reductions caused by benzarone, supporting selectivity of benzarone to EYA phosphatases.

8. Throughout the paper, WBs need molecular weight markers, and the legends should explicitly specify what “n” is. We need to know the number of cells quantified, and in how many independent

experiments the observations were made, and we also need to know how “n” was defined for statistical tests in each experiment. All the antibodies used should be specified in the paper.

We have added molecular weight markers and “n” values to the figure legends throughout. Antibodies have been listed in the Methods section.

9. Line 157, the statement that “These data demonstrate that EYA4 and EYA1 function redundantly to promote the accurate completion of mitosis through the dephosphorylation of PLK1” is certainly too strong in my view. Without showing that Plk1, or a mutant thereof, can rescue the defects caused by EYA1/4 loss, we only have correlation, not causation. Does Y445F rescue the effect of EYA1/4 loss?

We have now performed rescue experiments in EYA4 depleted cells and shown that Y445F rescues the effects of EYA4 loss. These data have been added to Figure 5 G-J. We have also amended the text to more accurately reflect the relationship between EYA1 and EYA4.

10. Experiments of the type in Figure 5B are not compelling. Loss of H3S10 phosphorylation in a WB could be caused by either failure to enter mitosis, or a shortening in the duration of mitosis. Any defect in cell cycle progression outside mitosis will cause loss of H3S10 phosphorylation. As Plk1 is implicated in mitotic entry, this is more than a theoretical concern. Live imaging of mitosis, as in Figure 3, would be needed to substantiate this effect. The fact that Plk1 WT does not rescue Plk1 RNAi is also curious. Why is this?

We believe our initial rescue experiments did not show a clear rescue with WT PLK1 because we performed the knockdowns before the overexpression, which may have caused mitotic arrest to begin before adequate WT PLK1 expression levels were reached. We have now repeated the rescue experiments with exogenous expression preceding knockdown, using both western blotting and live cell analysis of microscopy as outputs. These data demonstrate partial rescue with PLK1 WT and a greater rescue with Y445F PLK1. These data are now included in Figure 5C-F and are discussed in the text.

11. A potential problem with the model is that EYA1/4 is proposed to interact with Plk1-pY445 through the PBD of Plk1. However, pY445 is proposed to reduce the affinity of the PBD for its binding partners. How then does EYA1/4 bind to, dephosphorylate, and activate Plk1? Does EYA1/4 have higher affinity for the PBD compared to other binding proteins, even when Y445 is phosphorylated? Is pS128 really a PBD-binding site (see point 2 and 3)?

Our mutant data and structural simulations indicate that pY445 reduces the ability of the PBD to interact with other proteins without completely abolishing the interactions. While dephosphorylation of pY445 by EYA1/4 increases the PBDs affinity for protein-protein interactions, many other factors are likely to contribute. For instance, in the case of the EYA-PLK1 interaction, EYA levels, localization, and phosphorylation status of the putative PDS could locally or temporally overcome a less favourable PBD structure. This might suggest a threshold model of PLK1 activation by EYA4/1. Discussion of these ideas has been included in the revised manuscript.

Minor points

1. Line 40. I would say Cdk1-CycB is the master regulator of mitosis, not Plk1.

We have amended the text accordingly.

2. Line 81/Figure S1F. It would be good to show an alignment of the EYA sequences with the PDS consensus.

Alignments of the EYA sequences have been included in Figure S1F.

3. Separation, not seperation in Figure 1G.

Corrected.

4. Classifying as “1 or 2 centrosomes” in Figure 3 is confusing. The model seems to be that EYA1/4 affects centrosome separation, not duplication. Perhaps “1 or 2 centrosome-containing foci”, or similar, would be better.

Amended as suggested.

5. Please specify how mitotic duration and mitotic death were determined in Figure 3.

We have added a more thorough description to the Methods section.

6. The proteins and text inside the cells in Figure 6 are too small to read. Also, S128 does not have very obvious sequence features of a Cdk site, so saying “possibly” a Cdk site perhaps would be better than “likely”.

With the addition of new data, the model is now presented in Figure 7. We have removed the description of S128 as a “likely” Cdk site and redrawn the model for clarity.

7. It is somewhat surprising that Caron et al. Science Signaling (2016) 9, rs14 doi:10.1126/scisignal.aah3525 is not mentioned as context, but not essential.

This paper is now cited in the context of pY217.

Reviewer #3 (Remarks to the Author):

In the presented manuscript, authors report that function of a mitotic protein kinase PLK1 is regulated by tyrosine phosphorylation. Searching for potential substrates of a tyrosine phosphatase EYA1, they performed proximity biotinylation and identified PLK1 and several other centrosomal proteins as interacting partners of EYA1. They observed that knock down of EYA1 increased phosphorylation of PLK1 at tyrosine residues and using mass spectrometry, they identified Y445 as a major modified residue. In addition, depletion of EYA1 increased the level of PLK1 modification at pT210 that is used as proxy of PLK1 activation. Depletion of EYA1 (and its relative EYA4) caused mitotic defects similar as inhibition of PLK1 (including impaired centrosomal maturation and separation). Authors concluded that inactive PLK1 is phosphorylated at Y445 in G2 and EYA1-dependent dephosphorylation leads to activation of PLK1 at mitotic entry. Mechanisms controlling PLK1 activation are incompletely understood and also function of tyrosine phosphorylation in cell cycle is largely unexplored. Mechanism described in this manuscript is novel and surely could be of interest in the field. Most of the presented data is convincing but several additional experiments need to be done to allow the main conclusion that PLK1 acts downstream of EYA1/4. Addressing the issues detailed below would also significantly strengthen the manuscript.

Major points

1. Figure 3 shows that EYA1 and EYA4 depletion causes several mitotic defects that resemble inhibition of PLK1. Authors state that EYA4 and EYA1 function redundantly to promote the accurate completion of mitosis through the dephosphorylation of PLK1. However, this conclusion is not supported by the presented data. For this, authors need to show that the mitotic defects are rescued by expression of a PLK1 mutant that cannot be phosphorylated at Y445 or by constitutively active PLK1-T210D.

We have now performed rescue experiments using the phospho defective Y445F mutant of PLK1, and found that it does rescue EYA4 depletion better than PLK1 WT. These data are now included in Figure 5C-F. Additionally, we have removed the description of EYA4 and EYA1 function as “redundant”.

2. Based on results in Figure 2, authors conclude that EYA1 and EYA4 promote PLK1 activation. This might be the case but several issues need to be clarified. It is unclear how (and why) they quantified pT210 PLK1 signal at kinetochores (Fig. 2A)? As PLK1 and EYA1 colocalize at centrosomes, they should better quantify pT210 PLK1 signal at centrosomes rather than kinetochores?

Thank you for this comment. Quantification of pT210 was performed in the nuclei in general (not specifically at kinetochores). The reason for this was twofold: (i) while PLK1 activation is first observed at centrosomes, active PLK1 is fairly short-lived there, and migrates to and accumulates in the nuclei upon mitotic entry. Therefore, nuclear pT210 in mitotic cells provides a readout of overall PLK1 activation that has been initiated in G2 (and promoted at centrosomes by EYAs), (ii) the antibody we used to detect pT210 (Ab39058) produces an almost exclusively nuclear signal. To further demonstrate a reduction in PLK1 pT210 following EYA4/1 depletion, we have now repeated this experiment in HeLa cells using an alternative antibody (CAT# 9062 Cell Signalling), which stains both in the cytoplasm and in the nuclei (see also response to Reviewer #2, point 4). Staining with 9062 was combined with our PLK1 antibody such that per nuclei pT210/PLK1 ratios could be determined, supporting the changes in PLK1 pT210 levels occurring at a per-molecule level. These data are now included in Figure 2A-C of the revised manuscript, and the previous data generated with Ab39058 have been moved to Figure S2.

Figures 2B-D show decreased PLK1 pT210 signal upon depletion or inhibition of EYA1 or after expression of S128A EYA1. However, it seems that there is a reduction of pH3-S10 signal in all these conditions suggesting that observed PLK1 modification at T210 might be caused by decreased proportion of mitotic cells. Authors should evaluate PLK1-pT210 signal in mitotic cells collect by mitotic shake off. Benzarone also reduced the expression level of PLK1 in Fig. 2C and therefore authors should plot pT210/PLK1 ratio rather than pT210 level.

These experiments were done by mitotic shake off. This has been clarified in the Methods section of the revised manuscript. While we agree that there are some slight differences in the H3S10 staining across the blots, in all cases these changes mirror the slight changes in total H3, indicative of the fraction of mitotic cells being similar across treatments. EYA4 WT and the S128A mutant were also used in an IF experiment to measure changes in PLK1 pT210 in unsynchronized mitotic cells (Figure S2D), with these results supporting those produced by western blotting.

3. The staining of pT210 in the T loop of PLK1 might be influenced by modification of the close by residue Y217 that was reported as target of cAbl by Yang et al., 2017. Therefore, authors should formally demonstrate that PLK1 activity is decreased upon manipulation with EYA1. For instance, they can measure the activity of PLK1 immunopurified from mitotic cells by in vitro kinase assay.

To address this point, we used immunofluorescence to assess changes to the PLK1 substrates pS46 TCTP, pS133 CycB, pS198 CDC25C, *in vivo*. The cellular intensities of all three substrates decreased following EYA4 depletion, consistent with our conclusions. These data have been included in Figure 2D-G of the revised manuscript.

4. Fig. 1G – specificity of the staining with EYA4 antibody should be validated (for example by siRNA)

We have included siRNA verification of EYA4 antibody staining in Figure 1G.

5. The authors should elaborate more on the kinase responsible for Y445 modification. cAbl has previously been implicated in activation of PLK1 by phosphorylating Y217, Y425 and Y445 (Yang et al., 2017). Therefore, there seems to be a conflict with the model proposed here according to which phosphorylation of Y445 keeps PLK1 inactive during G2.

Yang et al., 2017 show changes in PLK1 pT210; however, these are attributed to pY425 (with pY425 being suggested to support PLK1 activation - the opposite of our proposed function of pY445). It is possible that pY445 and pY425 are both phosphorylated by cAbl, and that selective dephosphorylation by EYA4/1 favours the pro-activation effects of dephosphorylated Y445. However, our molecular dynamics simulation results suggest that pY425 does not influence PBD structure, suggesting that if pY425 does alter PLK1 activation, it likely does so via a different mechanism. We have included discussion of the contribution of cAbl to tyrosine phosphorylation and the interplay between the various phosphosites in the revised manuscript.

Minor points

1. EYA depletion increased pTyr signal of PLK1 purified from G2 cells but the signal was reduced later in mitosis (Fig 1D). Does it mean that another phosphatase targets PLK1 in mitosis? Why is no change of pTyr levels of PLK1 observed in control depleted cells?

It is possible that a different phosphatase takes over once mitotic entry has occurred. The reason that pY shows up in G2 may be attributed to knock down of EYA4 and/or there being higher levels of phosphorylation (potentially by cAbl). Tyrosine phosphorylation is difficult to detect on a western blot, so changes in phosphorylation may need to be above a certain threshold to be detected.

2. The statement that colocalization between EYA4 and PLK1 was strongest in cells prior to centrosome separation (Figure 1G) needs to be supported by quantification. Is there also some colocalization in the nucleus? Are these confocal or wide field images?

We have now quantified EYA4 intensity at PLK1 centrosomal foci and found that when PLK1 foci were near one another, EYA4 intensity was greater than when PLK1 foci were far apart. These data have been added to Figure S1G.

We do not consistently see nuclear colocalisations. In Figure 1, the intensity has been adjusted to show the centrosomal staining of EYA4, causing oversaturation of the nuclear signal, and potentially giving the appearance of some nuclear colocalization. Less saturated images show minimal nuclear colocalisations (see below). All the images were taken on a widefield fluorescent microscope, as described in the Methods section.

From new Fig 1G

3. Authors should consider reformulation of the statement that EYA1 and EYA4 are functionally redundant. In that case, depletion of individual phosphatase should not yield any phenotype.

We have amended the text to more accurately reflect the relationship between EYA1 and EYA4.

4. In Fig. 2D, the EYA4-S128A and Ydef mutants have clearly less cells in mitosis and therefore comparison of S128A and S128D is not informative. Instead, authors should evaluate statistical significance between WT and S128D where the amount of mitotic cells is comparable. Fig S2D shows comparison between WT and S128A in non-arrested cells but this is irrelevant as S128A will likely decrease the proportion of mitotic cells.

We agree that there are some slight differences in the H3S10 staining across the blots; however, as the H3 staining tracks with H3S10, we believe that the proportion of cells in mitosis is similar, and that any differences are attributed to inconsistencies in loading or protein transfer. Given that we evaluated the ratio of pT210/PLK1, we believe our comparisons are still reliable. Statistical significance was computed for all comparisons, and the differences shown are those that reached the significance threshold.

We disagree that the comparisons made in Fig S2D (now Fig S3A) are irrelevant. Even if there were differences in the proportion of mitotic cells, the measurements were only made in H3S10 positive mitotic cells, so all the cells evaluated were in mitosis.

5. Authors state that S128A has a dominant negative effect (Fig. 3K). I have hard time to understand how this could work if S128A does not bind PLK1. Instead, I would expect that pYdef mutant should have DN effect, but this is not the case as cells progress normally.

We have now summarized the observations in the manuscript without describing these relationships as being dominant negative because, as you suggest, the relationships may not be straightforward.

We don't know why Ydef does not increase mitotic duration; however, it does appear to have a moderate effect on the fraction of mitotic cell death events (Figure 3L).

6. Is there significant difference between pYdef mutant and the WT in Fig. 3L?

No, it is not significant.

7. What does Y445 scramble mean? It does not resemble the motif at all, so it should be called control pY peptide but not Y445

Y445 scramble has the same amino acid content as pY445 (the sequence was randomized). This has been clarified in the Methods section.

8. Figure S5A shows that Y445F mutant is strongly phosphorylated at other tyrosine residues. Does this pY signal respond to depletion of EYA1? How does Y217F mutant perform in this assay?

In this experiment, the Y445F mutant was expressed at higher levels than PLK1 WT, making the relative effect of EYA4 depletion comparable, but the absolute difference between PLK1 WT and Y445F not. It is possible that Y445F causes greater phosphorylation at other sites, and this has now been discussed in the revised manuscript. We have not investigated the effects of depleting EYA1 on the Y445F mutant or a Y217F mutant. While we acknowledge that these experiments would be interesting, we feel that they fall beyond the immediate scope of this manuscript.

Reviewer #4 (Remarks to the Author):

In the manuscript, Nelson et al. identified EYA4 as a key player for regulating PLK1 phosphorylation status on Y445, and its crucial role in regulating mitosis. Overall, it is an interesting and well written manuscript. The authors applied molecular dynamics simulations to suggest the potential molecular mechanism related to the effect of the phosphorylation. I have focused my review on those MD simulations. The applied methods and methodology are mainly sufficient and sufficiently described. This being said, the conclusion made from the presented results appear a bit thin, and I have the following concerns:

1) The timescale of the conducted simulations is short; thus, it is questionable if there is enough time for the proper system relaxation with the modified (phosphorylated) system to describe its biologically relevant configuration / conformational ensembles. In the manuscript, the authors point out that Y445F mutant was found to be hyperactive. I would find more confidence with the simulations (and their timescale) to describe the biologically relevant events/motions if simulation results of Y445F agree with the other presented results (and reflects correctly to the biological observations). The next points below are also of high importance in providing confidence on the simulation observations.

We have performed simulations of the Y445F mutant complexed with the phosphopeptide and found that the binding free energy is similar to that of the non-phosphorylated complex and significantly more negative than pY445. This agrees with the experimental observations. The results for the Y445 mutant have been included in the main text and Table S2 of the revised manuscript.

2) In figure 5D, an interaction between R507–pY445 is shown. Please provide information of the observed frequency of this R507–pY445 interaction in the simulations. Which interactions R507 display in the non-phosphorylated system? Furthermore, please demonstrate that the simulations are not biased due to the starting configuration of the pY445 system i.e. there is no impact with the close proximity of the R507–pY445 in the start conformation of the simulations (see the next point related to this). What is the role of the other important residues (close to Y445) in the dynamics of phosphorylated and non-phosphorylated systems e.g. Q452; E460; Y462; F409; D429 (e.g. their interactions)?

We have included the frequency of the R507–pY445 interaction and the other hydrogen-bonding partners of R507 in the WT.

R507 and Y455 are physically close to each other in many PDB structures and not just in 1UMW. Hence, we do not think it is necessary to demonstrate that our simulations are not biased towards the formation of hydrogen bonds between R507 and pY445. It can be argued that if these residues are not close enough in the starting structure, it slows down the convergence of the simulations. We may have to wait very long for the interaction to occur.

We have looked at the other interactions of R507 and pY445 and found that they form a hydrogen bonding network that also includes D429 and N446. This network does not exist in the non-phosphorylated state, as R507 does not interact with Y445. We have included this new observation in the main text of the revised manuscript.

3) The structure (1UMW) used in the simulations is not the best quality and there is no electron density data deposited with it to the PDB. It is also not mentioned, which chain of the structure was used in the simulations. There are better quality structures with electron densities available (important especially for R507 and the neighbouring residues), which illustrate also different conformations for these loop residues, especially R507 (e.g. PDB IDs: 3BZI; 3CL5; 3FVH; 3P2Z etc.). Would a simulation with a different starting conformation (structure) result in the same observations?

1UMW has good resolution ($<2\text{\AA}$) and accurately reflects the actual binding complexes formed by the PDB as it is bound to a consensus peptide. Other structures either contain shortened peptides (3FVH and 3P2Z) or are of poorer resolution (3BZI and 3C5L). As mentioned previously, R507 and Y455 are physically close to each other in many PDB structures, including the structures suggested by the reviewer. There are other PDB structures (e.g. 2OJX) with similar conformation for R507 as 1UMW. We do not expect significantly different observations when a different starting structure is used for simulation. We have now included information of the chains selected for simulation in the Methods section.

4) Figure 5E, F: Is this an average RMSF value from all the simulated replicas? Are there differences among replicas? Display the SD with the average to demonstrate that there is a difference between the systems (and confidence on the difference between the systems).

The authors could point out that there is observed disorder in the loop in some of the PLK1 structures (e.g. 2OCQ; 2OJX; 3P2W), which is nicely aligning with their RMSF data.

The RMSF shown is averaged over all the replicas. We have added close-ups of the 502-506 region with error bars included to the new Figure S6A-B. They show that there is a significant difference in flexibility for the residues in this region between the phosphorylated and non-phosphorylated systems. We have also pointed out that this region is associated with a high degree of disorder and is unresolved in some crystal structures in the main text.

5) It is rather well understood that the MMGBSA free energy estimates are highly suggestive. The authors should point out this limitation for the applied methodology in the manuscript. Also, there is no experimental values to compare here and their validity cannot be evaluated (see point 1 related to this below).

We have added this limitation of MM/GBSA to the main text.

Other points

1) A biophysical assay (e.g. ITC, SPR) demonstrating the discrepancy of binding affinities between Y445 and pY445 (PBD–substrate) would be a highly valuable addition to the manuscript.

We agree that this would be a valuable addition. However, we do not have the expertise or the resources to complete these experiments and it would require adding additional collaborators to the manuscript. We have therefore decided to forgo these experiments and have attempted to be as clear as possible about the inherent limitations of our results.

2) Fig 5D & Fig 6 labels in the figure are way too small to be easily seen. Please improve the readability.

Figure 5D has been replaced with a more detailed image with larger text (now Figure 6B). Figure 6 has been replaced with a simpler model with larger text (now Figure 7).

3) The authors do not state how the simulation data is made readily accessible for the readers. There are easy options, such as public repositories to upload the data (e.g. Zenodo). The out-dated statement “simulation data is available from the authors upon sufficient request” is not sufficient here.

We have uploaded our simulation data to Zenodo and included the doi in the Methods section.

4) The authors have conducted simulations with 4 replicas, although the general recommendations are a minimum of 5–10 replicas. (Knapp et al. J. Chem. Theory Comput. 2018, 14, 12, 6127–6138)

We have performed one more run for each system, for a total of five replicas.

5) In the methods, it is not well explained how the how the four replica simulations are independent.

We have added this information to the Methods section.

6) It is unclear why pY425 systems underwent slightly longer simulations, and therefore a different “time” is compared in MMGBSA with it to the other systems.

We have added the explanation to the Methods section.

7) Is there a specific reason that the authors were not using ff19SB force field?

Our entire simulation workflow was set up in AMBER18, which does not have ff19SB available.

8) Line 223: “The binding free energy of the interaction...”. Write more clearly which interaction is referred here.

We have clarified the interaction to be between the protein and peptide.

Reviewer #5 (Remarks to the Author):

Nelson et al. report a multi-faceted study that describes a novel signaling pathway in which EYA4 and EYA1 regulate PLK1 through the dephosphorylation of pY445 during G2. Starting with proximity proteomics data to interrogate the EYA4 interactome, they identify PLK1 among other centrosome

protein interactors, and proceed to biochemically characterize its phosphatase activity on Y445 and how this regulates mitotic progression via PLK1 activation. The authors note that this is the first report of its PLK1 Y445 dephosphorylation and functional significance. This appears to be a well-designed study with multiple lines of supporting evidence. I was asked to review this manuscript to provide perspective on the proteomics data, which is largely sound but requires addition of minor details as noted below.

1.) The authors need to provide proximity proteomics data as a spreadsheet (e.g., Excel file, .csv, .txt) in addition to the pdf format they currently have. Data difficult to examine and evaluate in the pdf format and pdfs are more challenging for informatics pipelines that may be used later to further explore this data.

Yes, and sorry about this. We had trouble getting excel sheets to upload on the Nature Commun. Platform, which was why we used the PDFs. The spreadsheets are now available.

2.) In the Methods, can the authors please include the number of entries present in the UP000005640_9606 FASTA file? Does it include all the isoforms of the human proteome? Are there any isoforms of EYA4, EYA1, or PLK1 present in the database that would complicate data analysis?

In addition to UP000005640_9606, UP000005640_9606_additional.fasta, which contains additional protein isoforms, was used in BioID and pY-IP experiments. This has now been added to the Methods section.

Isoforms for EYA4 (F2Z2Y1;O95677;O95677-2;O95677-4;E9PLN6;O95677-5;E7ESD5;O95677-3;E7EQM5;A6NCB9;F8WB53;Q99502-3;Q99502-2;Q99502;E7ETN2;O00167-3;O00167-2;O00167) and PLK1 (P53350;I3L2H5;I3L387;I3L309) were present within the data (EYA1 is not a part of the proteomic results). To our knowledge they present no complications to the analysis. PRM analysis only included UP000005640_9606.

The UP000005640_9606 FASTA file has 223,281 entries.

The UP000005640_9606_additional.fasta file has 546,697 entries.

As we were unable to find entry number listings for these files, the entries were instead counted in excel using =COUNTIF(A:A, ">*"). Entry counts for each FASTA file have been added to the Methods section.

3.) I may have missed it, but how many biological replicates were performed for the proximity proteomics experiment? I see replicates noted for the pTyr site-specific PRM experiments, but I did not see a replicate number that was used for statistics in the first LC-MS/MS experiment.

The proximity proteomics experiment was performed on 4 biological replicates. This is noted in the legend of Figure 1.

4.) Why does Supp Fig. 1e only use two replicates for data analysis when the Methods report that experiments were done in triplicate or quadruplicate? Or maybe I am misreading which replicates were used for pTyr site analysis versus pY IP-MS. Regardless, a clearer denotation of experiment design would be helpful. Also, PLK1 fold change is on the edge of significance in the volcano plot from the pY IP-MS data and would not be considered significant using the requirements they established for the proximity proteomics ($FC > 2$). The authors clearly have follow up data to support their conclusions, so this does not change the outcome of the manuscript... but knowing the authors

interpretation of a PLK1 pTyr barely making their significance threshold with not much fold change would be a useful addition.

Apologies for the confusion. This experiment was designed as n=3, however the HPLC column clogged after n=2 and we ended up losing the last rep. The Methods sections was referring to the pTyr site analysis. We have amended the text for clarity.

The PLK1 fold change was small in the pY IP experiment; however, it was large enough to produce a clear difference by western blot. Additionally, as there are several pY sites on PLK1 (perhaps even more than we were able to detect via MS), it is likely that a subtle effect in the overall pY signal is still meaningful at the level of individual residues. The pY IP-MS experiments were performed before we knew that EYA protein function was predominantly mitotic and before focusing on PLK1. Therefore, this experiment was done in asynchronous cells, and the modest change in PLK1 pY in this experiment may reflect this experimental design choice.

5.) If performing a gene set enrichment on subcellular location terms from all medium confidence and greater EYA4 interactors, is the centrosome/spindle among the top hits, or do EYA4 interactors share multiple subcellular locations? The authors logic on targeting centrosome or spindle proteins for the interactors is sound, but I am curious if this subcellular location is a dominant feature for EYA4 in these experiments. This would lend evidence to a “professional role” of EYA proteins in the centrosome, or it could indicate that there are several other subcellular locations that could be explored in subsequent work.

This is a good point. When we perform gene set enrichment for cellular component terms, we do indeed find that several terms related to centrosomes (“spindle pole”, “centriole”, “centrosome”, “microtubule organising center”) are enriched among EYA4 interactors. By strength of association these terms rank 8, 9, 14 and 15 out of 28 total enriched terms. Several of the other terms are related to transcription and RNA which may reflect the other functions of EYA4 as a transcriptional coactivator. Gene set enrichment data has been added to Table S1.

6.) Can the authors include the methods used for the pilot experiment performed to get the list of (pTyr) peptide targets for PRM? How confident are they that the phosphosite is Y instead of S or T in those peptides? This seems like a key point to address to make sure it is the pTyr site related to EYA phosphatase activity. CAnnotated spectra showing clear pTyr modification and no S or T mods would be highly conclusive for this.

Pilot experiment methods have been added to the Methods section. Additionally, greater detail about target ion selection has been added to the Results section. Example spectra showing clear pTyr modification have also been added to Figure S4.

7.) It appears the Skyline file used for analysis is not included in the PRIDE submissions that have the raw files, but it is possible I missed it. Can the authors include that file or verify that it is currently there? Also, I know they mentioned in the Methods that they followed guidelines presented in the Skyline tutorial, but if they could state some of these metrics (dotp scores, number of fragments used, etc), that would be very helpful to evaluate their strategy. Similarly, results files from the other LC-MS/MS experiments are missing in the PRIDE submissions, but they could be uploaded as supplemental spreadsheets as mentioned above if the authors do not prefer to add the files to the PRIDE submission.

Skyline results files have been uploaded to PRIDE and added to the revised manuscript, with greater detail provided for the description of the PRM results/methods. Additionally, results spreadsheets from the other LC-MS/MS experiments have been included in the Supplementary Data.

In the revised manuscript, we have included quantitative data only when at least 3 reps from each treatment group had all 6 fragment ions, and dotP scores ≥ 0.8 across all samples. While our results clearly demonstrate the existence of pY217, pY421, pY425, and pY445 in our cells, the streamlined data place a greater reliance on our mutation and functional experiments to demonstrate pY445 as the primary target of EYA4/1. Several rescue experiments as well as mutant resistance to EYA inhibition have now been added to Figure 5, which show conclusively that pY445 is an important target of the EYAs.

8.) Labels on the lanes in the WBs in Fig.1 could be better described. For example, does the lane (column) labeled “EYA4” mean EYA4 depletion? I assume so based on the text description and my interpretation, but having a description in the legend or a clearer label would help interpretation.

The figure has been amended for clarity.

9.) Line 140: “Total spindle defects increased modestly following EYA1 and EYA4 depletion” makes it sound like the combined condition, when really it is the individual conditions. Using “or” instead of “and”, or using “individual” in the same fashion that “combined” is used in the next sentence, could solve this confusion.

This has been corrected.

REVIEWERS' COMMENTS

Reviewer #1 (Remarks to the Author):

The authors have responded satisfactorily to my comments on the original manuscript. I recommend the revised version for publication.

Reviewer #2 (Remarks to the Author):

The authors have made serious efforts to address all of the various reviewers' comments. In particular, they now provide some data supporting that Plk1-Y445F mutants can rescue the effects of EYA1/4 loss, supporting their model (though would have been good to include some more specific assays of Plk1 function than mitotic duration and death). There are some minor points I think should be addressed prior to publication.

The claim on lines 172-173 ("These data demonstrate that EYA4 and EYA1 function to promote the accurate completion of mitosis through the dephosphorylation of PLK1") is still too strong at this point in paper, before the rescue experiments are described.

Line 221: delete "status"?

Figure 5: Parts 5G and 5I should have the same y axes. For 5G vs 5I and for 5H vs 5J, a statistical test should be carried out to compare Plk1 WT with Plk1 Y445F to see if there is reason to believe that the results are significantly different. (Note that lack of a statistical effect within Figure 5I is likely due simply to the spread of data for siEYA4). Similar for 5K: is the red line statistically different from controls? Depending on the result, the main text should be adjusted.

Reviewer #3 (Remarks to the Author):

In the revised manuscript, authors include new data that support their original conclusion that PLK1 activity is controlled by protein tyrosine phosphatases EYA1/4. In particular, they include validation of several antibodies, which was essential prerequisite for making any conclusions. They also include quantification of known substrates of PLK1 and show that their phosphorylation is reduced in mitotic cells upon depletion of EYA4. Finally, they edited the text and improved the discussion. Overall, authors successfully addressed all my points and I can now recommend the manuscript for publication.

Reviewer #4 (Remarks to the Author):

The authors have adequately addressed my concerns. I have only the following minor points related to the revised version:

I recommend for the clarity to change in line 274: "...phosphorylation of Y445 caused an increase in the binding free energy" to "...phosphorylation of Y445 resulted in a higher binding free energy".

The authors should cite examples of the structures instead of stating just "there are some"
Line 286: "resolved in some crystal structures "

In figure texts related to RMSF data (6 and S6) text term "wt" is used instead of "WT".

Revised Fig 6 has an error in line 972, (e) and (f) should be (c) and (d).

It could be useful for the readers to cite Figure S6 also in the new Fig 6 text, as it these images are highly related.

Reviewer #5 (Remarks to the Author):

The authors have adequately addressed my comments, and I have no further questions to address prior to publication.

REVIEWERS' COMMENTS

Reviewer #1

The authors have responded satisfactorily to my comments on the original manuscript. I recommend the revised version for publication.

Reviewer #2

The authors have made serious efforts to address all of the various reviewers' comments. In particular, they now provide some data supporting that Plk1-Y445F mutants can rescue the effects of EYA1/4 loss, supporting their model (though would have been good to include some more specific assays of Plk1 function than mitotic duration and death). There are some minor points I think should be addressed prior to publication.

The claim on lines 172-173 ("These data demonstrate that EYA4 and EYA1 function to promote the accurate completion of mitosis through the dephosphorylation of PLK1") is still too strong at this point in paper, before the rescue experiments are described.

We have altered this line to be more conservative.

Line 221: delete "status"?

We have deleted the word status.

Figure 5: Parts 5G and 5I should have the same y axes. For 5G vs 5I and for 5H vs 5J, a statistical test should be carried out to compare Plk1 WT with Plk1 Y445F to see if there is reason to believe that the results are significantly different. (Note that lack of a statistical effect within Figure 5I is likely due simply to the spread of data for siEYA4). Similar for 5K: is the red line statistically different from controls? Depending on the result, the main text should be adjusted.

The axis ranges of 5G and 5I have been made the same. T tests have now been carried out between PLK1 WT and PLK1 Y445F in figures 5G and 5I as well as 5H and 5J. While we did not observe a difference in the absolute level of mitotic cell death, we did find that EYA4 depletion resulted in a statistically longer mitotic duration in WT PLK1 overexpressing cells compared to Y445F PLK1 overexpressing cells. These comparisons have been added to the text. Additionally, we performed an ANOVA to compare the data in 5K and have found that the red line (Y445F) is statistically different from the control. We have included this comparison in the text.

Reviewer #3:

In the revised manuscript, authors include new data that support their original conclusion that PLK1 activity is controlled by protein tyrosine phosphatases EYA1/4. In particular, they include validation of several antibodies, which was essential prerequisite for making any conclusions. They also include quantification of known substrates of PLK1 and show that their phosphorylation is reduced in mitotic cells upon depletion of EYA4. Finally, they edited the text and improved the discussion. Overall, authors successfully addressed all my points and I can now recommend the manuscript for publication.

Reviewer #4 (Remarks to the Author):

The authors have adequately addressed my concerns. I have only the following minor points related to the revised version:

I recommend for the clarity to change in line 274: "...phosphorylation of Y445 caused an increase in the binding free energy" to "...phosphorylation of Y445 resulted in a higher binding free energy".

This has been edited in accordance with the suggestion.

The authors should cite examples of the structures instead of stating just "there are some"
Line 286: "resolved in some crystal structures "

We have added several citations to support this claim.

In figure texts related to RMSF data (6 and S6) text term "wt" is used instead of "WT".

This has been edited in accordance with the suggestion.

Revised Fig 6 has an error in line 972, (e) and (f) should be (c) and (d).

We have corrected this error.

It could be useful for the readers to cite Figure S6 also in the new Fig 6 text, as it these images are highly related.

Supplementary Figure 6 is already referred to in the text pertaining to Figure 6 in the revised manuscript:

"...particularly at residues 502-506, which is a region normally associated with a high degree of disorder and is not resolved in several crystal structures (Figure 6C-D, Supplementary Figure 6A-B)^{42, 56-58}. Loss of flexibility in the pY445 simulation occurred irrespective of whether the PBD structure was bound to the model phosphopeptide (Figure 6C-D, Supplementary Figure 6A-B)."

Reviewer #5 (Remarks to the Author):

The authors have adequately addressed my comments, and I have no further questions to address prior to publication.